# A systems-based approach to uterine fibroids identifies differential splicing associated with abnormal uterine bleeding

Chen-Yi Wang[1,6], Martin Philpott[1,6], Darragh P O'Brien [2], Anne Ndungu[3], Jessica Malzahn[1],
Marina Maritati[1], Neelam Mehta[1], Vicki Gamble[1], Beatriz Martinez-Burgo [3], Sarah Bonham[2],
Roman Fischer [2], Kurtis Garbutt [3], Christian M. Becker [3], Sanjiv Manek[3], Adrian L. Harris [4],
Frank Sacher[5], Maik Obendorf[5], Nicole Schmidt[5], Jörg Müller [5], Thomas M. Zollner[5],
Krina T. Zondervan [3], Benedikt M. Kessler [2], Udo Oppermann [1] ✉ & Adam P. Cribbs [1] ✉

## Abstract

**Background** Uterine fibroids (UFs), benign tumours prevalent in up to 80% of women of reproductive age, are associated with significant morbidity, including abnormal uterine bleeding, pain and infertility. Despite identification of key genomic alterations in *MED12* and *HMGA2*, the pathogenic mechanisms underlying UFs and heavy menstrual bleeding (HMB) remain poorly understood.

**Methods** To correlate systematically genetic, transcriptional and proteomic phenotypes, we conducted an integrative multi-omic approach utilising targeted DNA sequencing, RNA sequencing and proteomic methodologies, encompassing fibroid, myometrium, and endometrium tissues from 91 patients.

**Results** In addition to confirming the presence of *MED12* mutations, we identify variants in *AHR* and *COL4A6*. Multi-omic analysis of endometrium identifies latent factors that correlate with HMB and fibroid presence with driver mutations of *MED12*, *AHR*, and *COL4A6*, which are associated with pathways involved in angiogenesis, extracellular matrix organisation and RNA splicing. We propose a model, supported by in vivo evidence, where altered signalling of *MED12*-mutated fibroids influences RNA transcript isoform expression in endometrium, potentially leading to abnormal uterine bleeding.

**Conclusions** This study presents a comprehensive integrative approach, revealing that genetic alterations in UF may influence endometrial function via signalling impacts on the RNA splicing mechanism. Our findings advance the understanding of complex molecular pathways in UF pathogenesis and UF-associated endometrial dysfunction, offering insights for targeted therapeutic development.

## Plain language summary

Uterine fibroids are common benign non-cancerous tumours that grow in the womb and affect many women, often causing pain, heavy menstrual bleeding and problems with fertility. Genes are made up of DNA and are inherited. They provide instructions for making proteins and RNA, other molecules within the body. It is known that certain genes are associated with people having fibroids, but how fibroids cause symptoms like heavy menstrual bleeding is still unclear. We examined fibroid and endometrial tissues from 91 women and looked at the DNA, protein and RNA present. We found changes in fibroid tissues and discovered that these changes may also affect nearby endometrial tissues, which line the womb. This can alter how genes and proteins are expressed and may explain why bleeding occurs. These findings provide new insight into how uterine fibroids affect the body and may help develop better treatments to manage symptoms and improve women's health in the future.

Human uterine fibroids (UF), also known as uterine leiomyoma, are benign tumours of the uterus that affect a large population of women of reproductive age. They are particularly prevalent in black women in the United States, with an incidence of approximately 80% for those aged between 35 and 49, compared to 70% in white women of the same age group[1]. UFs interfere with normal uterine function, and in more than half of cases can cause distressing symptoms such as heavy menstrual bleeding (HMB), pelvic pain, urinary incontinence, and/or infertility[2]. Despite the high prevalence of the condition, treatment options are hindered by the broad range of clinical manifestations. Symptomatic UFs are treated either by

---

therapeutic interventions due to UF-growth dependence on female sex-steroid hormones, including selective progesterone receptor modulators (SPRM) such as Ulipristal acetate (UPA) and gonadotropin-releasing hormone (GnRH) agonist therapy[3–5], or by surgery, including hystero-scopic/laparoscopic myomectomy, embolization, hysterectomy[6]. In the United States alone, UFs are cited to be the cause of over 50% of hysterectomies[7], and direct costs for their treatment have been estimated to be between \$4–9 billion annually[8]. Irregular heavy menstrual bleeding (HMB; or AUB, abnormal uterine bleeding) is the most common symptom, affecting up to 46% UF patients[9]. HMB significantly impacts quality of life as a result of concurrent pain, anaemia, mood swings, and potential social embarrassment[10–12]. Despite its prevalence, the molecular mechanism linking UFs to HMB remains poorly understood, limiting targeted treatment options[12].

Mutually exclusive driver mutations in the mediator subunit 12 (*MED12*)[13] and high-mobility group AT-hook 2 (*HMGA2*)[14] genes occur in ~90% of UF cases. Med12 forms part of the Mediator Complex, which regulates transcription initiation and elongation by RNA polymerase II[15], while Hmga2 protein binds to, and alters the structure of DNA, promoting assembly of protein complexes that regulate transcription[16]. Other genetic contributors to UF include inactivation of fumarate hydratase (*FH*), a key enzyme of the Krebs cycle that promotes hypoxia when mutated[17,18], and dysregulation of the aryl hydrocarbon receptor (*AHR*), which influences extracellular matrix (ECM) formation and TGF-β signalling[19]. Additionally, deletion of the collagen genes *COL4A5* and *COL4A6* has been linked to familial UF cases[14,20]. However, how these mutations contribute to the development of UFs and associated symptoms are not yet fully understood.

Several studies have investigated the UF mechanism, primarily using microarrays to compare myometrium and fibroid, although the sample size of these early studies was limited[21–28]. Recent studies, such as Mehine et al.[20] for example, analysed 60 UFs with different genetic drivers (e.g., *MED12* mutations, *HMGA2* rearrangements, *FH* inactivation), revealing distinct pathway alterations in Wnt, prolactin, and IGF-1 (insulin-like growth factor 1) signalling. Proteomic approaches, despite small sample sizes of cohorts, have highlighted roles for apoptosis, inflammation, and cytokine regulation in the development of UFs. Collectively, these studies suggest UF development is linked to ECM, WNT-β-catenin and TGF-β3 signalling pathways[29–32].

In this study, we applied multi-omic approach of endometrium, myometrium and fibroid tissues from 73 UF and 18 non-UF patients to investigate the molecular mechanism underlying UF pathology and associated HMB. We identified key genomic alterations that provide insight into UF development. Integration of multi-omic factor analyses highlight the contribution of ECM dynamics and RNA splicing to UF-associated endometrial dysfunction. Differential transcript usage and single-cell transcriptomic profiling consistently point to aberrant TGF-β signalling and its role in modulating alternative splicing in the UF-affected endometrium. Our study provides insights into the molecular mechanism underlying uterine fibroid (UF), particularly in relation to heavy menstrual bleeding (HMB).

## Methods
### Patient samples and tissue collection
Fibroid, myometrium, pseudocapsule and endometrium tissues were collected from 137 donors undergoing hysterectomy, myomectomy or TransCervical Resection of Fibroids (TCRF) at the John Radcliffe Hospital, Oxford, in accordance with ENDOX study guidelines (09/H0604/58). All experimental protocols were approved by the local Research Ethics Committee (National Health Services (NHS) Research (NRES) Committee South Central-Oxford). Informed written consent was provided by patients participating in the study. In all cases, UF diagnosis was confirmed surgically and by histology. HMB status and use of hormone therapy was established from clinical notes and donor questionnaires. Menstrual cycle phase was determined by histopathology of the endometrium. Tissue samples were collected immediately after surgery, snap frozen in liquid nitrogen, and stored at −80 °C. The majority of the fibroid samples analysed were from the central region. However, pseudocapsule tissue was available in a limited number of patients, and where present, it was included in the study. Samples collected by TCRF tended to be of poor quality and yielded little or no endometrium, as did myomectomies, and surgeries performed by morcellation could not be reliably separated into individual tissue types. Overall, tissues from 91 donors were retained for this study and deemed suitable for this study.

### SureSelect targeted sequencing
DNA for SureSelect assays and SNP arrays was purified from fresh frozen samples stored at −80 °C using a PureLink Genomic DNA Kit (Invitrogen) according to the manufacturer's instructions for mammalian tissue. Eluted DNA was quantified by NanoPhotometer (Implen) and stored at −20 °C until further use. Approximately 100 ng of each DNA sample was used to create Illumina sequencing libraries using a NEBNext Ultra II FS DNA Library Prep Kit (New England Biolabs (NEB), E7805S). After PCR amplification with index primers, targeted DNAs were captured and enriched by SureSelect XT HS Target Enrichment Kit ILM Hyb Module according to the manufacturer's instructions (Supplementary Data 5, Agilent). Indexed libraries were quantitated by high-sensitivity DNA ScreenTape assay for TapeStation (Agilent), pooled at equimolar concentration, and sequenced on a NextSeq 500 to an average of ~8 million reads/sample. Reads were initially assessed for quality using FastQ Screen v0.14.0, FastQC v0.11.9 and MultiQC v1.5.dev0. Raw reads of each sample were mapped to hg38 using BWA v0.7.17 and merged into a single bam file. For SNPs and small insertions/deletions (indels), variant calling was performed using mpileup provided in bcftools v1.9[33] using human genome GRCh38, with default Bayesian genotype likelihood-based models and the parameters of minimum mapping quality as 20 and minimum base quality as 30, to detect variants. A likelihood ratio test was used to infer the probability of a variant at each site, and the QUAL score (phred-scaled *p*-value for the null hypothesis of no variant) was used to assess confidence. Sites with QUAL ≥ 50 (corresponding to 99.999% confidence) were considered as candidate variants. Variant annotation, effect prediction and associated phenotypes were performed by SnpEff[34] and Ensembl Variant Effect Predictor[35].

### Bulk RNA-sequencing
Tissue samples stored at −80 °C were cryomilled with Trizol without allowing the tissue to thaw. Briefly, one stainless steel end cap was inserted into a polycarbonate cylinder and precooled in liquid nitrogen along with the other cap and impactor. On dry ice, the impactor, 1.6 mL of Trizol, and the tissue sample were added to the cylinder, which was capped and placed in the cryomill. The procedure was performed for 3 cycles of 2 min. Once completed, samples were transferred to a 50 mL centrifuge tube pre-chilled on dry ice. When processing multiple samples, tubes were kept on dry ice or stored at −80 °C prior to downstream batch processing. Sample tubes were placed in a 37 °C water bath until thawed, vortex mixed, aliquoted into 1.5 mL centrifuge tubes and stored at −80 °C if not proceeding immediately to RNA extraction. RNA extraction was performed using a Direct-zol RNA miniprep kit (Zymo Research) and on-column DNAse I digest, according to the manufacturer's instructions. Eluted RNA was quantified by Nano-Photometer (Implen), quality checked by high-sensitivity RNA ScreenTape assay for TapeStation (Agilent), and stored at −80 °C until further use. RIN values generally ranged between 3 to 5, typical of tissue samples, but suggesting some 3' bias would be observed in the RNAseq.

Approximately 100 ng of each RNA sample was used to create Illumina sequencing libraries using a NEBNext Ultra II Directional RNA Library Prep Kit for Illumina with NEBNext Poly(A) mRNA Magnetic Isolation Module (New England Biolabs) according to the manufacturer's instructions. Indexed libraries were quantitated by high sensitivity DNA Screen-Tape assay for TapeStation (Agilent), pooled at equimolar concentration, and sequenced on a NextSeq 500 to an average of ~20 million reads/sample.

## Analysis of bulk RNA-sequencing

Reads were initially assessed for quality using FastQ Screen v0.14.0, FastQC v0.11.9 and MulitQC v1.5.dev0. Raw reads of each sample were then merged into a single file and pseudo-aligned to the human genome hg38 with Kallisto 0.46.0. The samples with alignment rate lower than 60% were excluded from downstream analysis. Using the count matrix produced by Kallisto, differential expression analysis was performed by DESeq2 v1.35.0[36] for comparisons with the clinical factors such as cycle phase, HMB, *MED12* status, and the technique factor like batch effect. Functional analysis, including gene set enrichment analysis (GSEA) and over-representation analysis (ORA) was done by R packages clusterProfiler 4.2.2[37,38]. For differential transcript usage analysis, raw reads of the samples were pseudo-aligned to gencode.v29.annotation.gtf by Kallisto, and the output abundance files were imported by tximport[39] and then analysed by DRIMSeq[40] and stageR[41]. Genes with differential transcript usage that passed the filter ($p < 0.05$ in DRIMSeq and then 5% overall FDR in stageR) were included in the final output of DTU analysis. Downstream analysis, including sequence alignment, conserved domain search, predicted protein structure of encoded protein isoforms, was performed using the tools msa[42], ragp[43], and AlphaFold 3[44], respectively.

## Uterine fibroid protein extraction

Frozen UF samples were cryomilled on liquid nitrogen in 1.6 mL of a lysis buffer comprising 6 M urea, 2 M Thiourea, 50% RIPA, 4% SDS, 100 mM DTT, and supplemented with protease and phosphatase inhibitors. To release protein bound to RNA and DNA, 1 µL of benzonase nuclease was added to 500 µL of each thawed sample and incubated on ice for 20 min. Due to the inherent toughness of the UF tissue samples, each was subjected to three rounds of bead beating for 2 min at 4 °C for maximum tissue disruption. Samples were then spun down for 5 min at 10,000 *g* and 4 °C. The supernatant was diluted (1:5) in water to achieve a final DTT concentration of 20 mM. Reduced samples were alkylated by adding IAA to a final concentration of 40 mM and incubated at room temperature for 1 hour in the dark. To remove SDS and other contaminants, all samples were subjected to a protein extraction procedure of alternating washes in methanol, chloroform and water. To maximise protein recovery, precipitated pellets were resuspended in 500 µL of 100 mM TEAB buffer, sonicated on ice for 5 min in a water bath, and vortexed at room temperature for 30 min. The protein content of each UF sample was then determined using a standard BCA assay.

## Sample digestion, clean-up, and TMT-labelling

Samples were digested in a 96-well format using the SMART Digest kit provided by Thermo Fisher Scientific. Briefly, 150 µg of each lyophilized UF sample was resuspended in 50 µL of 100 mM TEAB and added to 150 µL of the accompanying SMART Digest buffer. Frozen SMART Digest PCR strips containing immobilized trypsin beads were thawed and spun down at 1000 g for 1 min, and at 4 °C. Samples (200 µL) were transferred into the appropriate PCR tube and incubated on a heated shaker for 180 min at 70 °C and 1400 *g*. Upon completion, samples were spun down at 1000 g for 1 min. UF digests were cleaned-up with the aid of a vacuum manifold using the SOLAµ Solid-Phase Extraction (SPE) Plates provided with the kit. Samples were loaded in a 1:1 ratio (v:v) with 0.1% TFA, followed by one wash with 0.1% TFA. Peptides were eluted with 70% ACN into a 96-well collection plate and lyophilised to completion. For TMT-labelling, samples (~150 µg) were resuspended in 100 µL of 100 mM TEAB. Approximately 10% of each sample was removed for the preparation of global pooled samples. For this, two concentrations were prepared to be included in each TMT 10plex labelling reaction, one undiluted pool of all samples (1X Pool), and a five times diluted pool samples (5X Pool). Immediately before use, TMT label reagents were equilibrated to room temperature. To each 0.8 mg vial, 82 µL of anhydrous acetonitrile was added and the reagent allowed to dissolve for 5 min with occasional vortexing, before being gently centrifuged to gather the solution. For each TMT labelling reaction, 41 µL of the TMT label reagent was added to each 100 µL of UF sample. The reaction was allowed to proceed for 1 hour at room temperature before being quenched for 15 min with 8 µL of a 5% hydroxylamine solution. For each TMT 10plex experiment, an equivalent volume (140 µL) of sample was combined, resulting in a total protein amount of approximately 1.5 mg in a final volume of 1.4 mL. Each concatenated sample was desalted on a C18 solid-phase extraction cartridge (Sep-Pak Plus, Waters).

## High-pH reversed-phase pre-fractionation

Approximately 1.5 mg of digested TMT-labelled material was subjected to off-line high-pH reversed-phase pre-fractionation using the loading pump of a Dionex Ultimate 3000 HPLC with an automated fraction collector and a XBridge BEH C18 XP column (3 × 150 mm, 2.5 µm pore size, Waters no. 186006710). Peptides were separated over a 100 min gradient using two basic pH reversed-phase buffers (A: ammonium hydroxide in 100% water, pH 10; B: ammonium hydroxide in 90% acetonitrile, pH 10). The gradient consisted of a 12 min wash with 1% B, then increasing to 35% B over 60 min, with a further increase to 95% B over 8 min, followed by a 10 min wash at 95% B and a 10 min re-equilibration at 1% B. The flow rate was set to 200 µL/min, with fractions collected every 2 min throughout the run. In total, 50 fractions were collected over the run, but samples were concatenated down to a final of 10 fractions by combining every 10th sample. Each fraction was dried down and resuspended in 30 µL of 2% ACN:0.1% formic acid for analysis by LC–MS/MS.

## High performance Liquid Chromatography Tandem Mass Spectrometry (LC-MS/MS)

LC-MS/MS analysis was performed using a Dionex Ultimate 3000 nano-ultra high pressure reversed-phase chromatography system coupled on-line to a Q Exactive High Field (HF) mass spectrometer (Thermo Scientific). Samples were separated on an EASY-Spray PepMap RSLC C18 column (500 mm × 75 µm, 2 µm particle size; Thermo Scientific) over a 60 min gradient of 2–35% acetonitrile in 5% DMSO, 0.1% formic acid and at 250 nL/min. The mass spectrometer was operated in data-dependent mode for automated switching between MS and MS/MS acquisition. Full MS survey scans were acquired from $m/z$ 400–2000 at a resolution of 60,000 at $m/z$ 200 and the top 12 most abundant precursor ions were selected for HCD fragmentation. The resolution of MS2 fragment ion detection was also set to 60,000. Fractions were loaded with adjusted sample volumes to analyze ~1 µg on column.

## Proteomics Data Analysis

MS raw data were searched against the UniProtKB human sequence database (92,954 entries) and TMT 10plex quantitation performed using Proteome Discoverer software (v 2.3; Thermo Scientific). Search parameters were set to include carbamidomethyl (C) as a fixed modification, with TMT 6plex, oxidation (M), and deamidation (NQ) set as variable modifications. A maximum of 2 missed cleavages was allowed. TMT 10plex quantitation and data analysis were performed in Perseus (v1.6.0.2), resulting in the generation of hierarchical clustering, principal component analysis, and Volcano plots. For PCA analysis, samples underwent $\log_2$ transformation and all missing values were removed. This was then followed by median subtraction normalisation. For the generation of volcano plots, an identical processing workflow was used, but only 50% of the missing values were removed. The missing values that remained were imputed from the normal distribution (width 0.3, down shift 1.8). Differentially regulated proteins between groups of interest were subject to gene ontology and pathway enrichment analysis using STRINGdb (https://string-db.org/). Shortlisted targets were further assessed for their biological relevance and therapeutic potential in the treatment of UFs using TargetDB (https://pypi.org/project/targetDB/).

## Integration of transcriptomics and proteomics by Multi-Omics Factor Analysis (MOFA)

In addition of the metadata containing the clinical information related to donors, the log-normalized count matrices of transcriptomics and

proteomics (Supplementary Data 4) were used as the input data to MOFA[45,46]. In the proteomic data, features that contain more than 50% missing values were removed. MOFA is an unsupervised statistical method to integrate multiple modalities of omics data and to identify latent factors that capture sources of variation across datasets obtained from different platforms. The latent factors represent coordinated variation across data modalities, but do not inherently have predefined biological meanings. A MOFA object was prepared using default settings and trained under a slow convergence mode, with the number of factors suggested by the algorithm. The likelihood for both the transcriptomic and proteomic data was both inferred as Gaussian.

The MOFA model was trained in a Bayesian framework, which differs fundamentally from classical regression models that rely on p-values for inference. Instead of computing p-values to assess feature significance, MOFA applies sparsity-inducing priors and automatic relevance determination (ARD) that allow the model to estimate the relevance of each feature through posterior inference. In this context, feature loading weights represent the strength and direction of contribution of each feature to a given factor. Using sparsity in the weights, loading weight of many features are exactly zero, indicating their irrelevance to the factor, while only a subset of features has non-zero weights, meaningfully contributing to latent factors. Thus, the selection of relevant features is not based on statistical significance using classical regression, but on the magnitude of their contribution as inferred by the posterior distributions of the model.

For functional interpretation, gene set enrichment analysis (GSEA) was conducted using built-in function of MOFA with default setting, including "mean.diff" (difference in the average weight between fore- and back-ground genes) for gene set statistic, a parametric t-test, Benjamini-Hochberg procedure to adjust p-values factor-wise for multiple testing, and false discovery rate (FDR) threshold 0.1 for significant pathways. All features associated with factors were used as input. Pathways enriched in both omic layers were prioritised. Shared features associated with factors in both modalities, with absolute loading weights higher than a cut-off value of 0.3, were visualised using by STRINGdb. While this threshold is not derived from p-value, it serves as an interpretable cutoff to highlight features with stronger associations. Overrepresented pathways were analysed via the Enrichr database[47] using its R interface.

Known clinical, biological and technical covariates were correlated with MOFA-inferred factors to support interpretability. These included genotype information (e.g., *MED12* status and SNPs), fibroid occurrence, tissue type (e.g., UF or myometrium), menstrual cycle phase, HMB symptom, hormone treatment, and batch effects. While several latent factors showed associations with these known variables, other may reflect unknown sources of variation for future investigation.

## Nuclei preparation for single-cell RNAseq
A petri dish, 50 ml centrifuge tube, scalpel and forceps were precooled on dry ice before pseudocapsule samples were removed from −80 °C and placed in the petri dish. Typical sample sizes ranged from 100–500 mg. Tissue was cut into thin slices and transferred to centrifuge tubes. If processing multiple samples, cut tissue could be stored at −80 °C until use. Sample tubes were transferred to wet ice, 4 ml of ice-cold CST buffer (146 mM NaCl, 10 mM Tris-HCl pH 7.5, 1 mM CaCl$_2$, 1 mM MgCl$_2$, 0.5% CHAPS (w/v), 0.01% BSA (w/v), 4 μl/ml SUPERaseIN, 4 μl/ml RNasein Plus, 1 cOmplete protease inhibitor tablet (per 10 ml)) added and tubes placed on a rotator for 10 minutes at 4 °C. Samples were passed through 30 μm cell strainers (MACS SmartStrainer) into prechilled 15 ml collection tubes on ice. Sample tubes were rinsed with 2 ml ice cold PBS + 1% BSA, which was added to the cell strainer. Cell strainers were rinsed with an additional 2 ml ice-cold PBS + 1% BSA and collection tubes centrifuged at 500 g for 5 minutes at 4 °C. Supernatant was removed and pellet washed by resuspending in 10 ml ice cold PBS + 1% BSA, centrifugation at 500 *g* for 5 minutes at 4 °C, removal of supernatant and resuspension in 500 μl ice

cold PBS + 1% BSA. A subsample of the nuclei preparation was incubated with DAPI (1 μg/ml) for 5 minutes, added to a haemocytometer and counted under a fluorescent microscope. Concentration of the nuclei was adjusted to ~1,000 cells/μl and used as input for analysis by 10X Chromium single cell gene expression.

## Library preparation and Sequencing of single-cell RNA sequencing
Chromium single cell gene expression (10X Genomics) was performed using the Chromium Next GEM Single Cell 3′ GEM, Library & Gel Bead Kit v3.1, Chromium Next GEM Chip G Single Cell Kit and Single Index Kit T Set A according to the manufacturer's instructions, starting with 20,000 nuclei as input. Resulting libraries were quantitated by TapeStation (Agilent), pooled at equimolar concentration and sequenced (Novogene (UK) Ltd or Genewiz GmbH) on an Illumina NovaSeq 6000 using a S4 Reagent Kit v1.5 to give ~30,000 reads/cell.

## Analysis of Single cell RNA sequencing
Raw sequencing data (fastq files) were processed using the scflow workflows (https://github.com/Acribbs/scflow). The Kallisto BUS/BUStools (v0.39.3) workflow1 was implemented to pseudo-align the reads, with a K-mer size of 31 base pairs. Homo sapiens (human) genome assembly GRCh38 (hg38) was used to construct a reference transcriptome. Individual samples of single-nuclei or single cells were analyzed by the pipeline of quantnuclei or quantcells implemented in the scflow workflows, respectively. The output was converted to single-cell experiment objects[48] and then to Seurat objects (Seurat v4.0)[49]. Quality control and filtering were performed on the Seurat objects; any cell with a mitochondrial ratio higher than 0.1, or fewer than 300 features was removed. Doublets in the samples were detected using the R package scDblFinder[50] and removed in the scflow pipeline with Seurat clustering.

To integrate the endometrium samples with the published data, we first used the VST method provided by Seurat for variable gene selection and applied Harmony v1.04[51] for batch correction. Highly variable genes that account for cellular heterogeneity in each main cluster were used and cells were aligned using Harmony. For cell-cell communication, we applied CellChat (v1.4.0)[52] with input of two matrices, log log-normalized count matrix and a matrix of the cell label.

## THESC decidualization
The cell line T HESCs was received from ATCC (ATCC® CRL-4003™) certificated mycoplasma free. Cells were incubated in DMEDM/F-12 with bicarbonate and HEPES (Sigma Cat# D 2906) supplemented with 10% foetal bovine serum (FBS, Charcoal stripped F6765-500ML), puromycin (500 ng/ml), and 1% ITS Premix Universal Culture Supplement (Corning 354350). For the three-day experiment of decidualization, cells were seeded in 6-well plates for 40,000 cells per well and incubated overnight. At the next day (Day 0), the decidualization were induced by adding the following reagents into cell medium: Medroxyprogesterone 17-acetate (Sigma, M1629; final conc. 1.0 μM), E2 (estradiol, final conc. 10 nM; Sigma E1024), 8-Br cAMP (8-Bromoadenosine 3′,5′-cyclic monophosphate, final conc. 500 μM; Sigma B6386-100mg). In addition to stimulation for decidualization, cells were further treated with DMSO as mock, TGF-β (10 ng/ml; Millipore GF346) together with or without MEK inhibitor (BAY 1076672, 100 ng/ml) since Day 0, depending on the experimental design. Cells were harvested on Day 3 using Direct-zol RNA MiniPrep kit (Cambridge Bioscience, R2052).

## Library construction and sequencing of Nanopore long-read sequencing
50 ng RNA of each sample were reverse transcribed and barcoded by using the PCR-cDNA barcoding kit (SQK-PCB111.24) and NEBNext Companion Module (NEB E7180L). Libraries were then sequenced on the Nanopore PromethION platform.

### Analysis of long read sequencing to identify transcript isoforms

Base calling of fast5 files was done by Guppy (https://github.com/asadprodhan/GPU-accelerated-guppy-basecalling) and converted to fastq format. Fastq files were processed by the pipeline (pipeline_count) implemented in the workflow TallyTriN (https://github.com/cribbslab/TallyTriN/tree/main) and raw count matrix for PCA analysis was then generated. Reads of each sample were then aligned to hg38 genome by Minimap2[53] with - - MD flag enabled and output as SAM format. The SAM file of each sample was processed by TALON[54] v5.0 and Swan[55] v2.0, using default settings, and gencode.v29.annotation.gtf as reference for isoform-level analysis (transcript switching genes and transcripts which are not in the gencode database due to alternative splicing) and visualisation.

### Mice

Female mice (Balb/c) (purpose-bred animals, Janvier Labs) aged ~9 weeks were housed according to the EU guideline 2010/63 EU. The study (study code: A0384/09) was approved by the German animal welfare authorities (LAGeSo, Berlin).

### Mouse model of menstruation and treatment regimens

The experimental model of menstruation in mice was adapted from established protocols[56]. Female Balb/c mice were acclimatized to the animal facility for one week before being trained in animal handling for at least one week prior to inclusion in the study. Cage enrichment, such as nesting material and hiding structures, was provided throughout the study to improve the well-being of the animals. Mice underwent bilateral ovariectomy, with analgesia provided before and after surgery by administering tramadol (1 mg/ml) orally via drinking water. Overall, the study is regarded as mildly burdensome, with no standard need for additional analgesic treatment.

One week post-surgery, mice received subcutaneous injections of 100 ng 17α-estradiol (E2) dissolved in a 1:9 ethanol to peanut oil solution for three consecutive days. After a three-day interval, a subcutaneous silastic implant delivering progesterone (P4, internal source[57]; 0.5 mg P4/day) was inserted dorsally. Concurrently, 5 ng of E2 was administered daily for three consecutive days. On the final day of E2 treatment, 50 μl of sesame oil was injected into one uterine horn to induce decidualization. Four days later, the P4 implant was removed to trigger progesterone withdrawal.

To assess menstrual-like bleeding, tampon-like cotton pads (4–4.8 mm in diameter) were inserted into the vagina of mice at the time of P4 withdrawal. Mice were fitted with paper collars to prevent the removal of the pads. Tampons were replaced twice daily, and samples from each mouse were collected individually. Blood volume was quantified using the alkaline hematine method[58]. Briefly, tampons were first dried at room temperature and then immersed in 1000 ml of 5% sodium hydroxide (NaOH, w/v) overnight under rotation at room temperature to dissolve haem chromogen. The optical density of the haem-containing eluates was measured at 546 nm using an ELISA plate reader. Blood volume contained in cotton swabs was measured based on a standard curve prepared from venous blood.

Seventy-two hours after P4 withdrawal, mice were euthanized, under deep terminal anaesthesia with isoflurane (>3%), by terminal blood collection from the vena cava. Uterine tissues were collected, weighed, and processed for further analyses. All surgical interventions were conducted under isoflurane-induced anaesthesia, with pain prevention provided by tramadol treatment. Notably, no animals in this study experienced unexpected severe events or required rescue analgesic treatment, and no animals were excluded from the experiment or final analysis. Mice were randomly allocated to placebo and treatment groups, and the treatment of the animals was not blinded, as the primary readout, the quantitative ex vivo measurement of blood loss, was performed blinded to the operator.

### Treatment in the mouse model

Groups ($n = 10$) were treated with either the MEK inhibitor (BAY MEKi, cpd 26[59], Bayer AG, Germany) at doses of 0.5 mg/kg/d p.o. or with the ACVR1 inhibitor (TP-0184, Toledo Pharmaceuticals, USA) at doses of 15 mg/kg/d p.o. dissolved in N-methyl-2-pyrrolidone (NMP)/ polyethylene glycol 400 (PEG400) (1/9) (d0-d15) in a volume of 5 ml/kg. Controls were treated with vehicle alone qd/p.o.

### Statistics and reproducibility

Transcriptomics and proteomics data matrices used as input for the multi-omics factor analysis (MOFA) are provided in the Supplementary Data 4. Prior to analysis, proteomics data was filtered to retain features detected in at least 50% of samples and then normalised and log transformed. Transcriptomics data were normalized and variance-stabilising transformed using DESeq2. The MOFA model was trained with default parameters, including Gaussian likelihoods, sparsity priors like spikeslab_weights and ard_weights, and a slow convergence setting (corresponding to an ELBO tolerance of 5e-8). The number of latent factors was inferred based on MOFA model performance.

Clinical information (Supplementary Data 3) and genotype information (Supplementary Fig. 2) were used to investigate the biological relevance of each factor. Unlike classical regression models that rely on frequentist statistical significance (e.g., p-values), MOFA operates within a Bayesian framework that estimates the relevance of each feature using sparsity-inducing priors. Most features have zero contribution (loading weight is zero), while a subset of features with non-zero loading weights meaningfully contribute to the latent factors. As a result, MOFA does not calculate p-values for feature-factor associations. Instead, the magnitude of the loading weight of each feature on a latent factor indicates its importance and direction of contribution. While some Bayesian models report posterior inclusion probabilities to quantify confidence in feature inclusion, MOFA identifies relevant features based on their inferred weights. In this study, features with absolute loading weight higher than a cut-off value of 0.3 in both modalities were considered highly biologically relevant and selected for visualisation in STRINGdb.

For Fig. 5c, two independent in vivo experiments were conducted to investigate the effects of TP-0184 (an ACVR inhibitor) and BAY-533 (a MEK inhibitor). Data analysis (Supplementary Data 6) was performed using GraphPad Prism 10 software. For comparisons between the two respective groups, statistical significance was assessed using a one-sided Student's $t$-test ($p < 0.05$; $****p < 0.0001$). Based on extensive prior experience with this model, the data particularly regarding bleeding as the primary endpoint are considered robust. Due to ethical constraints and in agreement with the established reliability of the model, repetition of the animal experiments was not approved by the local regulatory authorities.

## Results
### Clinical features of the cohort

A total of 91 patients, predominantly European population, undergoing hysterectomy, myomectomy or trans-cervical resection of fibroids (TCRF) were recruited (Supplementary Fig. 1; Supplementary Data 1). The majority had uterine fibroids (UFs), while 18 non-UF patients served as a comparative cohort, though they were not considered as healthy controls. These patients underwent surgery for conditions including endometriosis, adenomyosis, ovarian cysts or cervical neoplasia. Heavy menstrual bleeding (HMB) status was determined via patient questionnaires and clinical records, with 33 donors classified as HMB (Supplementary Fig. 1). As hormone treatment influences HMB symptoms, patients undergoing such treatment at the time of surgery were assumed to have therapeutic intervention. Menstrual cycle phase was primarily determined histologically, with clinical notes and hormone levels used when histology was unavailable. Notably, 35 patients had inactive endometrium due to hormone treatment. The collected tissues encompassed distinct uterine compartments, including endometrium, myometrium, fibroid, as well as pseudocapsule, a vasculature-rich region that surrounds the tumour, which is not formed in all fibroids (Supplementary Data 1).

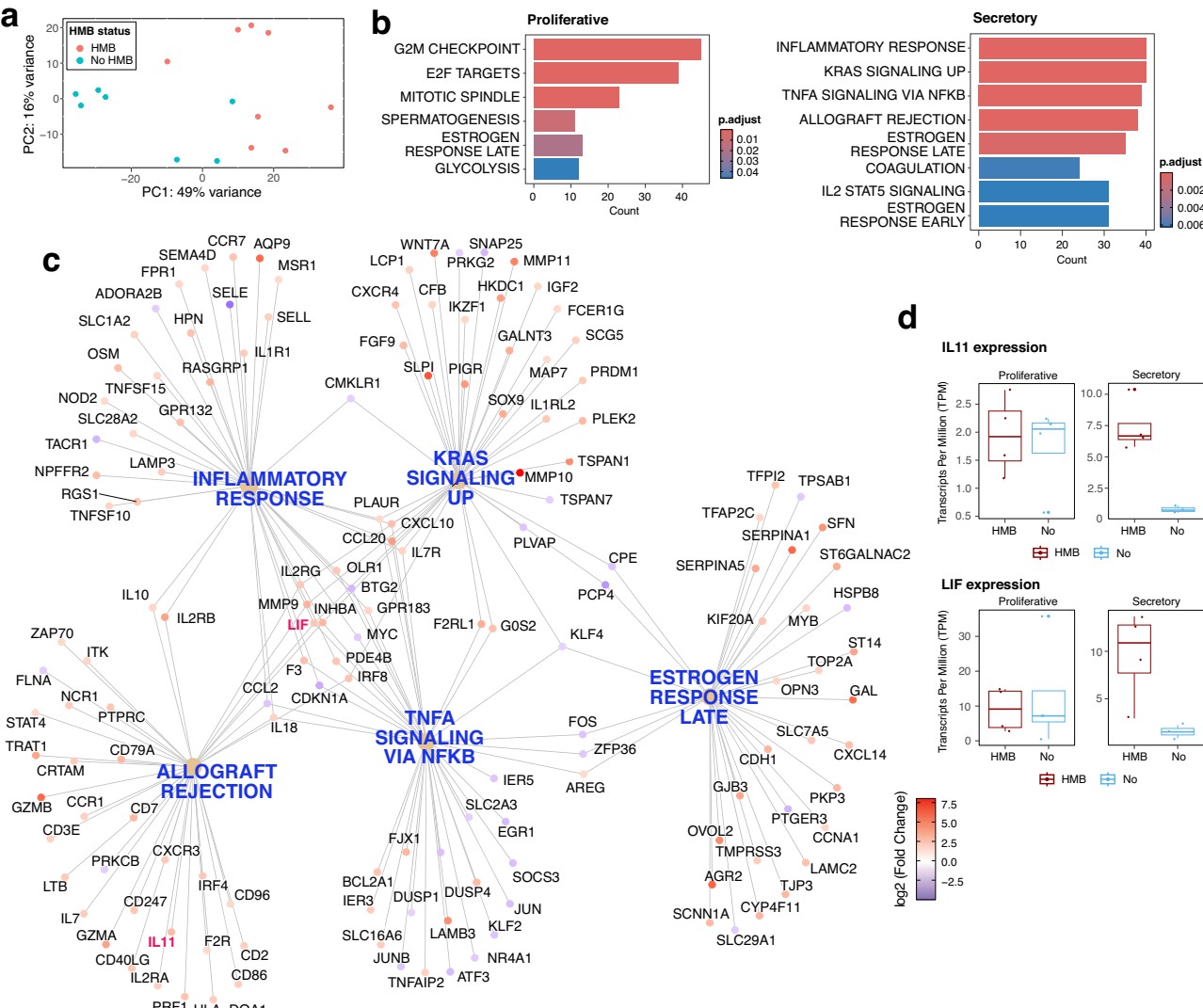

**Fig. 1 | Differentially expressed genes and enriched pathways in active endometrium from UF patients. a** Principal component analysis (PCA) plot showing clustering of endometrium samples from UF patients with or without heavy menstrual bleeding (HMB) symptoms (*n* = 15; proliferative phase: 4 HMB and 4 non-HMB; secretory phase: 4 HMB and 3 non-HMB). **b** Bar plots of pathways enriched in HMB endometrium, identified by gene set enrichment analysis (GSEA). Pathways are ranked by adjusted p-value, shown in a gradient of blue to red. Left and right panels correspond to the proliferative and secretory phase, respectively. **c** Network visualisation of differentially expressed genes (DEGs; absolute log₂FC ≥ 1.5, padj < 0.05) in secretory HMB endometrium, depicting the linkages of gene functions and pathway associations. **d** Boxplots of *IL11* and *LIF* expression in the proliferative (left panel) and secretory (right) phases of the endometrium.

## Genomic insights into UF pathology

To investigate UF-associated genetic alterations, we performed targeted sequencing of key UF driver genes, including *MED12, HMGA2, FH, COL4A5/6, HMGA1*[60], *RAD51B*[14], *AHR*[61], *CAPRIN1*[61], *CUX1*[62], *DCN*[61] and *PCOLCE*[63] candidate genes. The variant calling analysis focused on single nucleotide polymorphism (SNPs) and short indels. A likelihood ratio test was used to infer the probability of a variant at each site, and sites with QUAL score (phred-scaled p-value for the null hypothesis of no variant) ≥ 50 (corresponding to 99.999% confidence) were considered as candidate variants (Supplementary Data 2). Among 73 fibroids, 39.7% harboured *MED12* variants, which are canonical UF mutations in intron 1 and exon 2, with other *MED12* variants having minimal functional impact (Supplementary Fig. 2a). Furthermore, we identified mutation hotspots in *COL4A6*, *AHR* and *CUX1*, including in-frame insertion-deletion and frameshift mutations in *COL4A6* exon 24, and missense variants in *AHR* exon 10 and *CUX1* exon 16 (Supplementary Fig. 2b, upper, middle and bottom panel, respectively).

## Differential gene expression in UF HMB endometrium

To investigate gene expression profiles in the endometrium of UF patients with HMB, we applied bulk RNA sequencing and performed differential gene expression analysis using DESeq2. Principal component analysis (PCA) (Fig. 1a) exhibited distinct separation between HMB and non-HMB patients with active menstrual cycle, along the PC1 and PC2 axes. Gene set enrichment analysis (GSEA), using a p-value cutoff of 0.05 and Benjamini–Hochberg (BH) adjustment for multiple testing, revealed that during the proliferative phase, the gene expression profile was dominated by cell cycle and mitotic processes (Fig. 1b, left panel), whereas during the secretory phase, immune-related pathways including inflammatory response and allograft rejection, as well as RAS signalling, were enriched (Fig. 1b, right panel; Fig. 1c). These findings are consistent with established roles of inflammatory processes and leucocyte trafficking in the endometrial physiology[64]. *IL11* and *LIF* for example, were significantly upregulated in HMB patients (log₂ fold change (log₂FC) > 1.0, adjusted p-value (padj) < 0.05), particularly in the secretory phase (Fig. 1c, d). Recombinant human

IL11 has shown > 50% reduction[65] (ClinicalTrials.gov ID: NCT00524342) in pictorial blood assessment chart (PBAC)[66], which is widely used to assess menstrual blood loss, implying a role for immune dysregulation on HMB.

## Multi-omic factor analyses identify dysregulation of ECM and RNA processing as key contributors to UF-associated HMB symptoms

To gain deeper insight into UF-associated HMB, we applied multi-omic factor analysis (MOFA)[45,46], an unsupervised method that integrate bulk transcriptomics, proteomics and genomics, and identifies latent factors that capture sources of variation across datasets obtained from different platforms. We examined whether the identified latent factors were associated with the known clinical, biological and technical variables including patient type (UF or non-UF patient), HMB status (with or without HMB symptom), mutations identified in fibroids, and sample batches. MOFA analysis of 31 endometrial samples (UF and non-UF patients; Supplementary Data 3-4) identified 7 latent factors (Fig. 2a, c). Factor 1 was significantly correlated with HMB and hormone treatment (padj < 0.01), suggesting a lasting impact of therapeutic interventions on endometrium function. Factor 2 was strongly associated with fibroid presence (padj < 0.001; Supplementary Fig. 3), indicating the influence of fibroid tissue on physiological functions of endometrium. Factor 7 was correlated with not only the presence but also genomic alterations of fibroid, including *MED12* UF mutations, *AHR* rs2066853 and *COL4A6* rs6622312, all of which were also correlated with HMB (padj < 0.05; Fig. 2a right & Fig. 2b, c).

The relevance of each feature to a latent factor is identified by MOFA via a Bayesian framework with sparsity-inducing priors. The contribution of each feature is inferred as a loading weight by posterior distribution of the MOFA model[45,46]. Loading weight of irrelevant features is exactly zero, while features with non-zero loading weights on a latent factor indicate the strength and direction of contribution. GSEA analysis of Factor 1-associated features revealed the enrichment in coagulation, angiogenesis and ECM organisation (false discovery rate (FDR) < 0.1) in both omics (Fig. 2d). Most of these features with stronger association with Factor 1 (absolute loading weight ≥ 0.3) were negatively association with HMB (Fig. 2e). For example *CD59*, whose genetic deficiency is linked to haemolytic anaemia and thrombosis[67], and angiogenin (*ANG*), an RNAase A superfamily member involved in neovascularization[68], were downregulated in HMB endometrium.

Pathway analysis of Factor 2 and Factor 7 identified enrichment of RNA processing and metabolic process, including mRNA splicing and RNA 3'-end processing (FDR < 0.1; Fig. 2f, g; Supplementary Fig. 3). These findings suggest UF-induced dysfunction of RNA homoeostasis and the subsequent aberrant splicing events in endometrium may be exacerbated by *MED12*, *AHR* or *COL4A6* variants in fibroids, potentially contributing to HMB symptom.

Integrated analysis of fibroid (n = 50) and myometrium (n = 41, including 31 UF and 10 non-UF patients) identified 6 latent factors (Supplementary Fig. 4). Factor 2, unaffected by batch effects, strongly correlated with tissue type (padj < 0.001; Supplementary Fig. 4a–c) and was associated with pathways related to ECM and collagen fibril organisation, angiotensin maturation, and hormone metabolic process (FDR < 0.1; Supplementary Fig. 4d, e). *UCHL1*, a ubiquitin C-terminal hydrolase involved in protein homoeostasis, was positively associated with fibroid tissue (Supplementary Fig. 4e) and has been implicated in promoting TGF-β signalling via stabilisation of the type I TGF-β receptor[69]. Higher level of *UCHL1* in UFs[70] may contribute to the elevated TGF-β signalling. These findings were further supported by 2D Annotation Enrichment analysis[71] (Supplementary Fig. 5), reinforcing the central role of ECM dysregulation in fibroid pathology. MOFA analysis of fibroid samples showed a noteworthy albeit weak correlation with *MED12* UF mutations, *AHR* rs2066853 and *COL4A6* rs6622312 at Factor 5 (padj < 0.01; Supplementary Fig. 6a–c). Enriched pathways by GSEA also highlighted ECM, collagen fibril organisation and angiogenesis, addressing the crucial role of ECM in UF pathology (FDR < 0.1), and indicating these variants may exacerbate ECM dysregulation (Supplementary Fig. 6d, e).

## Differential transcript usage reveals the role of TGF-β signalling and RNA processing in UF endometrium pathology

Our integrated analysis identified RNA processing and mRNA splicing as key molecular mechanisms underlying UF endometrium pathology. To examine transcript-level alterations in the endometrium of UF patients with HMB, we performed differential transcript usage (DTU) analysis on active endometrium samples, excluding those under therapeutic hormone treatment to minimise confounding effects. Using DRIMSeq for initial DTU detection (p-value < 0.05) and stageR for further statistical testing (overall false discovery rate (OFDR) < 0.05)), we identified 684 transcripts across 478 genes in differential transcript usage between HMB (n = 8) and non-HMB (n = 7) patients. Alternative transcript usage was observed in genes including *TGFBR2*, *ENG*, *NRP1*, *TBXA2R*, and *PDE1A*, which are involved in blood vessel morphogenesis and angiogenesis, prostaglandin synthesis and regulation, and calmodulin-mediated signalling, respectively (Fig. 3a). Pathway enrichment analysis highlighted processes related to vascular smooth muscle cell differentiation, peptide antigen assembly with MHC complexes, and ribosome biogenesis (Fig. 3b).

When comparing endometrial samples from patients with *MED12*-mutant fibroids (n = 10) and those with wild-type *MED12* (n = 5), 2,784 transcripts across 2134 genes (p-value < 0.05 and OFDR < 0.05) were identified. We observed DTU in genes involved in protein modification, stress-activated MAPK cascade, mRNA transport and RNA splicing such as *HNRNPR* and *HNRNPL* (Fig. 3c, d). Notably, TGF-β signalling emerged as a key pathway, with DTU analysis identifying alternative usage in *TGFBR1*, *TGFBR2* and *TGFBR3* in TGF-β receptor signalling pathway, as well as *ANGPT1* and *ANGPT2* in angiogenesis (Fig. 3c and e). These findings underscore the roles of TGF-β signalling in UF-induced dysfunctions in endometrium, particularly in the presence of *MED12* mutations. Given that SMAD3-mediated TGF-β signalling directly regulates alternative splicing[72,73], the observed DTU of TGF-β receptors may impact downstream signalling dynamics. A striking example is *TGFBR2*, which encodes two alternative spliced variants, TβR-II and TβRII-B, with distinct ligand-binding affinities. TβR-II, which binds TGF-β1/3, and TβRII-B, which binds TGF-β2[72,73]. Intriguingly, our analysis found that TβRII-B (ENST00000359013) was the dominant isoform in the endometrium of patients with HMB or with *MED12*-mutant fibroids (Fig. 3a and e). As shown in Supplementary Fig. 7, an additional peptide composed of 25 amino acid residues in the extracellular domain of TGF-βRII[74] alters TGF-βRII protein structure, suggesting a shift in TGF-β ligand specificity in this pathological context.

Most genes identified through DTU analysis did not exhibit differential expression at the gene level (padj < 0.05, absolute log2FC ≥ 1.5), with only a small subset overlapping between DTU and differential gene expression analysis. In addition to angiogenesis, DTU analysis identified genes involved in prostaglandin synthesis (*PTGES*, *PTGES2*, and *PTGFR*), progesterone signalling (*PGR*), and FGF signalling (*FGF7* and *FGFR2*). These findings further highlight transcript-level regulation as a crucial layer of molecular control in UF pathology and suggest that alternative splicing may contribute to UF-associated symptoms like HMB.

## Single-cell transcriptomic analysis reveals altered TGF-β signalling and ECM remodelling in UF endometrium –

The impact of fibroids on endometrial function has been reviewed by Ikhena and Bulun[75]. Elevated TGF-β3 secretion from fibroids is implicated in disrupting wound healing and coagulation pathways, potentially contributing to HMB[76]. To explore the molecular and cellular differences between UF and healthy endometrium, we applied single-cell RNAseq (sc-RNAseq) on secretory-phase endometrial samples from UF patients with HMB (n = 4), integrating them with healthy secretory-phase endometrium[77] (n = 5). Following batch correction, quality control and cell annotation (Supplementary Figs. 8, 9 and Fig. 4b), we identified 4 major cell types, further subdivided into 10 cell clusters (Fig. 4a, b), including lymphatic endothelial cells, macrophage, and dendritic cells, with notable differences of cell composition between normal and UF tissues (Supplementary Fig. 10).

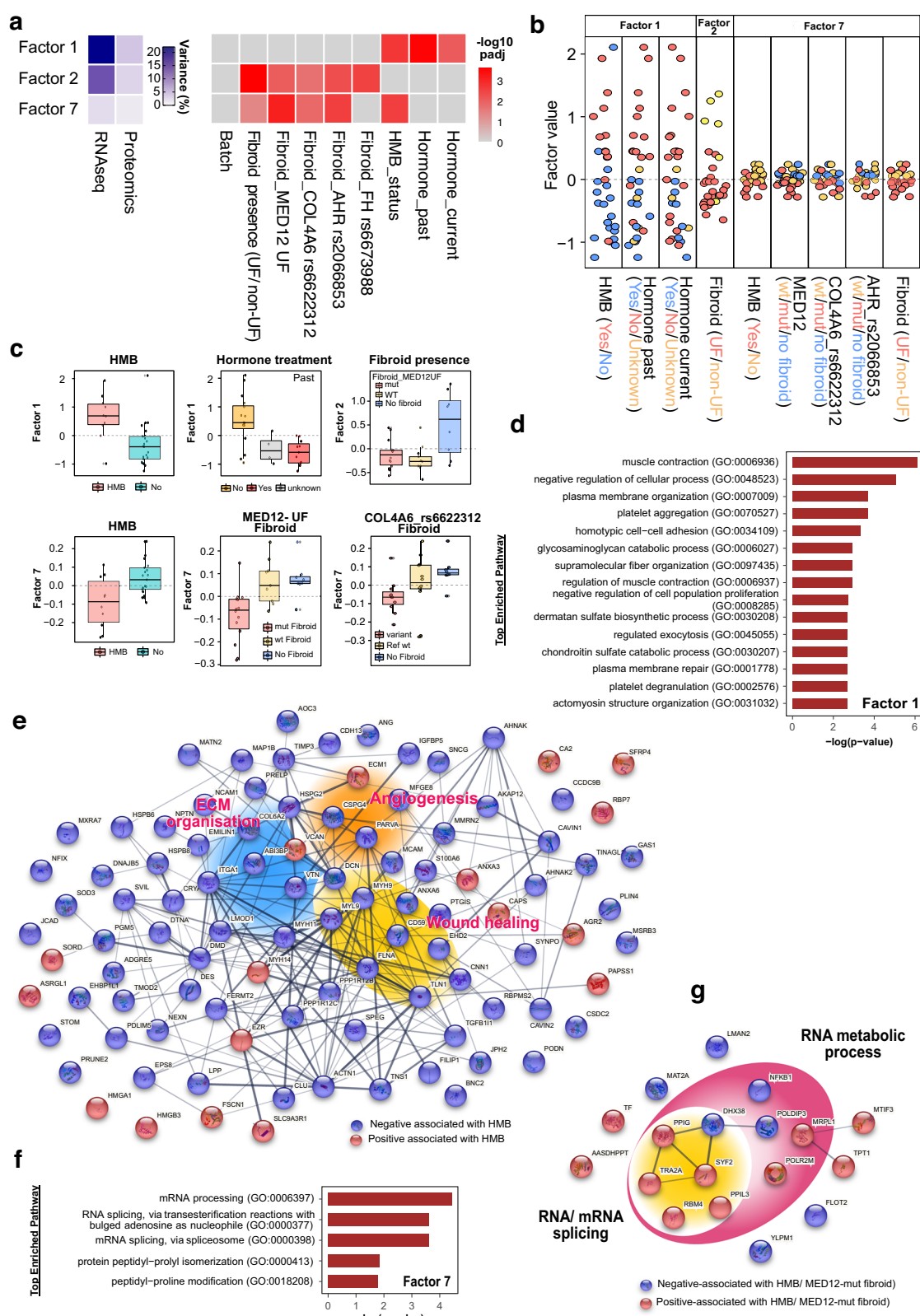

To investigate cellular communication networks, we performed CellChat[52] for ligand-receptor interaction analysis. We observed strikingly increased cross-talks between UF endometrial cell clusters, compared to healthy controls (Fig. 4c). Among the enriched signalling pathways (p-value < 0.05), signalling by TGF-β superfamily, was markedly upregulated (Fig. 4d), with higher expression of TGF-β-associated receptors, including *TGFBR1*, *TGFBR2*, *BMPR1A*, *BMPR1B*, *BMPR2* and *ACVR1* in UF endometrium (average log$_2$FC > 1.3, padj <0.05; Fig. 4e). Given the established elevation of TGF-β in fibroid[78,79], these results suggest that fibroid-derived TGF-β ligands may contribute to aberrant signalling in surrounding uterine tissues, potentially exacerbating HMB and ECM remodelling.

**Fig. 2 | Integrated analysis of endometrium using multi-omics, including transcriptomics, proteomics, and targeted genomic sequencing. a** Left: The relative contribution of the transcriptomic and proteomic datasets to MOFA-inferred factors, expressed as the percentage of explained variance, with intensity represented in blue. Right: Correlation of factor variance with clinical and genetic parameters, quantified by $-\log_{10}$ (adjusted p-value) and visualised in red. Parameters include experimental batches ($n = 3$), fibroid presence (UF vs non-UF), hormone treatment (past or current), heavy menstrual bleeding (HMB) status, and fibroid-associated mutations: canonical *MED12* UF mutations, *COL4A6* rs6622312, *AHR* rs2066853 and *FH* rs6673988. **b** Scatter plots illustrating the differentiation of samples based on key clinical and genetic parameters, including HMB (Yes, $n = 10$; No, $n = 21$), hormone past (prior hormone treatment: Yes, $n = 11$; No, $n = 16$; Unknown, $n = 4$), hormone current (treatment at time of surgery: Yes, $n = 8$; No, $n = 20$; Unknown, $n = 3$), fibroid (UF, $n = 23$; non-UF, $n = 8$), *MED12* UF mutations (wt, $n = 14$; mut, $n = 9$; non-fibroid, $n = 8$), *COL4A6* rs6622312 (wt, $n = 12$; mut, $n = 11$; non-fibroid,

$n = 8$), and *AHR* rs2066853 (wt, $n = 16$; mut, $n = 7$; non- fibroid, $n = 8$). MOFA factor values represent the relative positioning of samples, with larger absolute values indicating stronger associations. **c** Boxplots showing the distribution of sample groups across MOFA factors 1, 2, and 7, revealing variance within these factors. The centre line represents the median; boxes represent the interquartile range (IQR), and whiskers extend to 1.5 times of IQR. **d** Gene ontology (GO) enrichment analysis highlighting pathways of features contributing to Factor 1 in both omics (FDR < 0.1). **e** STRING network diagrams elucidating the interactions among features associated with Factor 1 in both modalities (absolute loading weight higher than 0.3). The loading weight of each feature was identified by MOFA using Bayesian framework and sparsity-induced priors, different from classical regression using p-values for significance. Only relevant features have non-zero loading weight. **f** GO enrichment analysis of features contributing to Factor 7 in both omics (FDR < 0.1). **g** STRING network diagrams of features associated with Factor 7 in both modalities (absolute loading weight higher than 0.3).

Apart from TGF-β receptors, *TGFB2* expression was also notably elevated in UF endometrium (average $\log_2 FC > 3.0$, padj <0.05), compared to healthy controls. As shown in Fig. 4f, TGF-β2-mediated signalling via the dimer of the type I TGF-β receptor (TGFBR1) and ACVR1 revealed that in healthy endometrium, signalling was primarily restricted to stromal clusters (p-value < 0.05), whereas in UF endometrium, it was widespread across multiple cell types, indicating differences in TGF-β signalling. In addition, abnormal signalling pathways, including effectors such as collagen, laminin and fibronectin (FN1) were observed in UF endometrium and myometrium, compared to normal tissues[77,80] (Supplementary Figs. 11, 12). This also underscores alterations in ECM composition and basement membrane architecture, which potentially compromise tissue homoeostasis and may contribute to UF-associated pathophysiology.

### TGF-β signalling in THESC cells induces alternative splicing

Our findings from bulk short-read RNAseq experiments suggested that TGF-β signalling induces alternative splicing changes in uterine tissues. To investigate the hypothesis that transcript isoform shifts are triggered by TGF-β in the endometrium, we treated the hTERT-immortalized human endometrial stromal cell line (THESC) with TGF-β during in vitro decidualization and monitored transcript-level changes using Nanopore long-read RNA sequencing (Fig. 5). This approach enabled precise determination of transcript isoforms. Consistent with a short-read (Illumina) THESC dataset, differentially expressed genes (padj < 0.05, absolute $\log_2 FC \geq 1.5$) were enriched in pathways related to cell cycle regulation and chromosome segregation (Supplementary Fig. 13). The pro-fibrotic effects of TGF-β are mediated through both SMAD-dependent and non-canonical MEK/ERK signalling pathways[81,82]. Prior studies have shown that blocking MEK/ERK can attenuate fibroid cell proliferation and ECM production, suggesting that ERK activation is required for certain TGF-β-mediated effects in fibroid pathology[83,84]. Given that aberrant ECM accumulation and dysregulated angiogenesis are key contributors to fibroid-associated HMB, we applied a MAPK/ERK kinase (MEK) inhibitor (MEKi)[59] to determine whether TGF-β-mediated signalling relevant to these processes was dependent on MEK/ERK activation.

To identify and quantify transcripts isoforms in the long-read dataset, we employed Talon[54] for transcript annotation and quantification, followed by Swan[55] for differential isoform expression analysis. Notably, TGF-β treatment during decidualization led to DTU events ($p < 0.05$) compared to DMSO, particularly in genes involved in mRNA processing and splicing, such as the hnRNP family[85–87] (*HNRNPA1*, *HNRNPA2B1*, *HNRNPC*, *HNRNPK*, *HNRNPR*, *HNRNPU*), RNA-binding proteins (*RBM4*, *RBM39*), VEGFA-VEGFR2 signalling pathways, and hereditary leiomyomatosis (Supplementary Fig. 14a). Similar pathways were enriched when comparing co-treatment with TGF-β and MEKi to TGF-β treatment alone during decidualization, indicating that TGF-β-driven transcriptome reprogramming is largely achieved through RNA metabolic process and mRNA splicing (Supplementary Fig. 14b).

As shown in Fig. 5, TGF-β altered transcript isoform ratios in multiple genes. For instance, *HNRNPA2B1* exhibited a shift from 100% *A2B1-202* to a 50:50 ratio of *A2B1-202* and *A2B1-206* upon TGF-β treatment (Fig. 5a, middle panel). Given that *A2B1-206* is an intron-retained, non-protein-coding transcript, this shift suggests potential downregulation of *HNRNPA2B1*. Similarly, we detected 10 *HNRNPC* transcript isoforms (Fig. 5a, bottom panel), including *HNRNPC-206* (ENST00000553444), a non-protein coding variant, while other isoforms encode structurally distinct proteins, suggesting functional changes due to transcript switching. These findings indicate that the functions of hnRNP family are regulated via alternative transcript usage, subsequently further influencing mRNA splicing and processing.

In addition to splicing-related genes, alternative splicing in ECM-associated genes was observed (Fig. 5b). Fibronectin-1 (FN1), a key ECM glycoprotein, mediates cell adhesion, integrin signalling, and growth factor binding (including TGF-β interactions)[88–90]. With distinct domain compositions, FN1 isoforms display different ligand-binding affinity, dimerization, solubility, and fibrillogenesis[88,89]. Among the *FN1* transcripts identified (Fig. 5b, upper panel), three are protein-coding. *FN1-208* encodes a 73 kDa N-terminal protein, *FN1-213* encodes a 121 kDa central/C-terminal protein, and *FN1-207* encodes a 239 kDa full-length isoform lacking EDA and EDB regions. These isoforms may exert differential effects in ECM organisation. Additionally, periostin (POSTN), a secreted ECM glycoprotein involved in fibrosis and tumour progression[91], exhibited alternative splicing between exon 17 and exon 21 (Fig. 5b, bottom panel), consistent with its differential expression in normal and diseased tissues[91].

To examine potential effects on HMB by blocking TGF-β or MAPK pathways, we further tested MEK and ACVR1 (TGF-β receptor) inhibition in an in vivo mouse menstruation model[56]. This system mimics primate menstrual cycles, where progesterone withdrawal induces menstrual-like bleeding in ovariectomised, hormone-primed mice. MEK or ACVR1 inhibition significantly reduced uterine bleeding (Fig. 5c), supporting the hypothesis that UF-associated growth factors affect endometrium physiology that potentially contributing to HMB.

Overall, our findings in both decidualized THESC cell line and endometrium from UF patients reveal that TGF-β signalling alters transcript usage in genes involved in mRNA splicing and ECM organisation. These alternative splicing events may underlie key pathological changes in UF, contributing to endometrial dysfunction and heavy menstrual bleeding.

### Discussion

The molecular mechanism linking UFs to HMB remains poorly understood, limiting targeted treatment options while current treatments primarily aim on reducing menstrual blood loss. UF growth is a female sex-steroid hormone-dependent process, accordingly therapeutic interventions for HMB have often focused on steroid hormones, oestrogen and progesterone, including selective progesterone receptor modulators (SPRM) such as Ulipristal acetate (UPA) and gonadotropin-releasing hormone (GnRH)

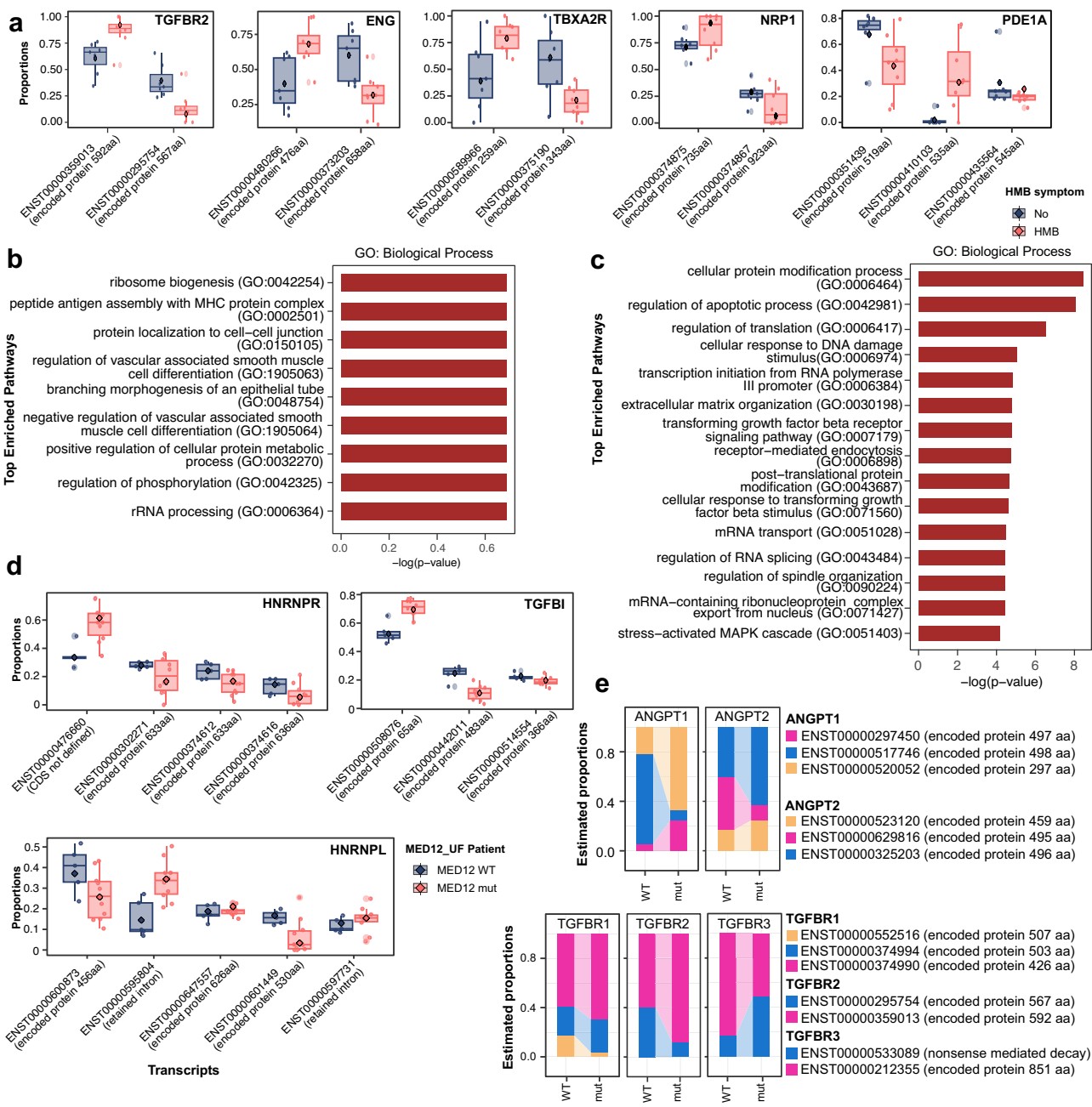

**Fig. 3 | Comparative analysis of transcript usage in active endometrium from patients with heavy menstrual bleeding or *MED12*-mutated fibroids. a** Boxplots displaying the expression of differentially used transcript variants in the active endometrium of UF patients with heavy menstrual bleeding (HMB, n = 8; coloured in pink-orange) compared to non-HMB patients (n = 7; grey). The centre line represents the median, while the lower and upper hinges correspond to the 25th and the 75th percentiles. **b**, **c** Bar plots of enriched pathways associated with genes exhibiting differential transcript usage, identified using DRIMSeq (*p*-value < 0.05)

and stageR (OFDR < 0.05). **b** Pathways enriched in HMB versus non-HMB endometrium. **c** Pathways enriched in endometrium from *MED12*-mutant versus *MED12* wild-type (WT) fibroid patients. **d** Boxplots showing the expression of differentially used transcript variants in active endometrium from *MED12*-mutant UF patients (n = 10; pink-orange) versus *MED12* WT (n = 5; grey). **e** Ribbon plots illustrating transcript usage shifts between *MED12* WT and *MED12*-mutant conditions, highlighting dynamic usage patterns across transcript variants of individual genes. Each transcript per gene is represented by a distinct colour.

agonist therapy[3–5]. Due to their side effect profiles, non-hormonal therapies that efficiently and safely target HMB are highly desirable. Our study provides insights into the molecular mechanism underlying uterine fibroid (UF), particularly in relation to heavy menstrual bleeding (HMB). By applying the multi-omics analysis of transcriptomics, proteomics, and genomics, in addition to single cell RNAseq (sc-RNAseq) analysis and differential transcript usage (DTU) analysis, we identified alternative transcript usage, TGF-β signalling and ECM dysregulation as key molecular

alterations that contribute to fibroid pathogenesis and endometrial dysfunction.

Our targeted sequencing approach reveals, in contrast to prior reports suggesting that *MED12* and *HMGA2* mutations are present in ~90% of UFs, a lower frequency of these mutations (<50% of cases) in our cohort. The reason for the discrepancy is unknown but the data may point to ethnic and regional differences in genomic aberrations found in UFs[92,93]. Instead, we observed a higher prevalence of *AHR* missense mutations, and *COL4A6*

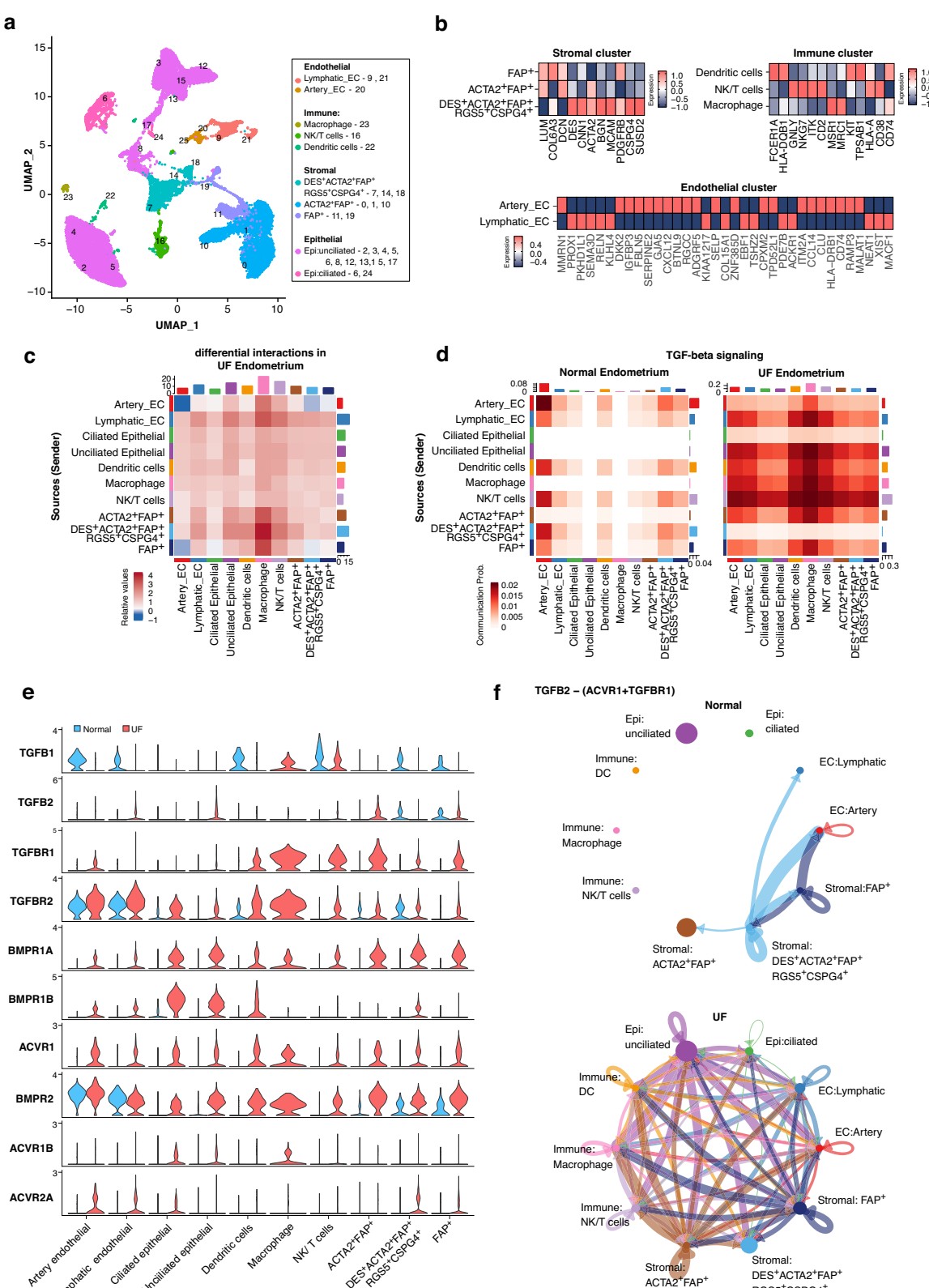

**Fig. 4 | Single-cell analysis of endometrium from UF patients with heavy menstrual bleeding compared to healthy controls. a** UMAP of the integrative single-cell dataset of UF ($n = 4$) and healthy endometrium ($n = 5$). Colours represent distinct cell subclusters within major cell types. **b** Heatmaps exhibiting average expression of canonical marker genes used for cell type annotation in stromal (upper left), immune (upper right) and endothelial (bottom) clusters. **c** Heatmap of differentially enriched cell-cell interactions in UF endometrium compared to healthy controls. Relative values of interaction strength is indicated by a gradient from blue (low) to red (high). **d** Heatmap displaying TGF-β signalling across cell clusters in normal (left) and UF (right) endometrium. **e** Violin plots illustrating the expression of ligands and receptors involved in TGF-β signalling in normal (blue) and UF (red) endometrium. **f** Circle plots showing inferred TGFB2-(ACVR1 + TGFBR1) signalling among different cell types in normal (top) and UF (bottom) endometrium.

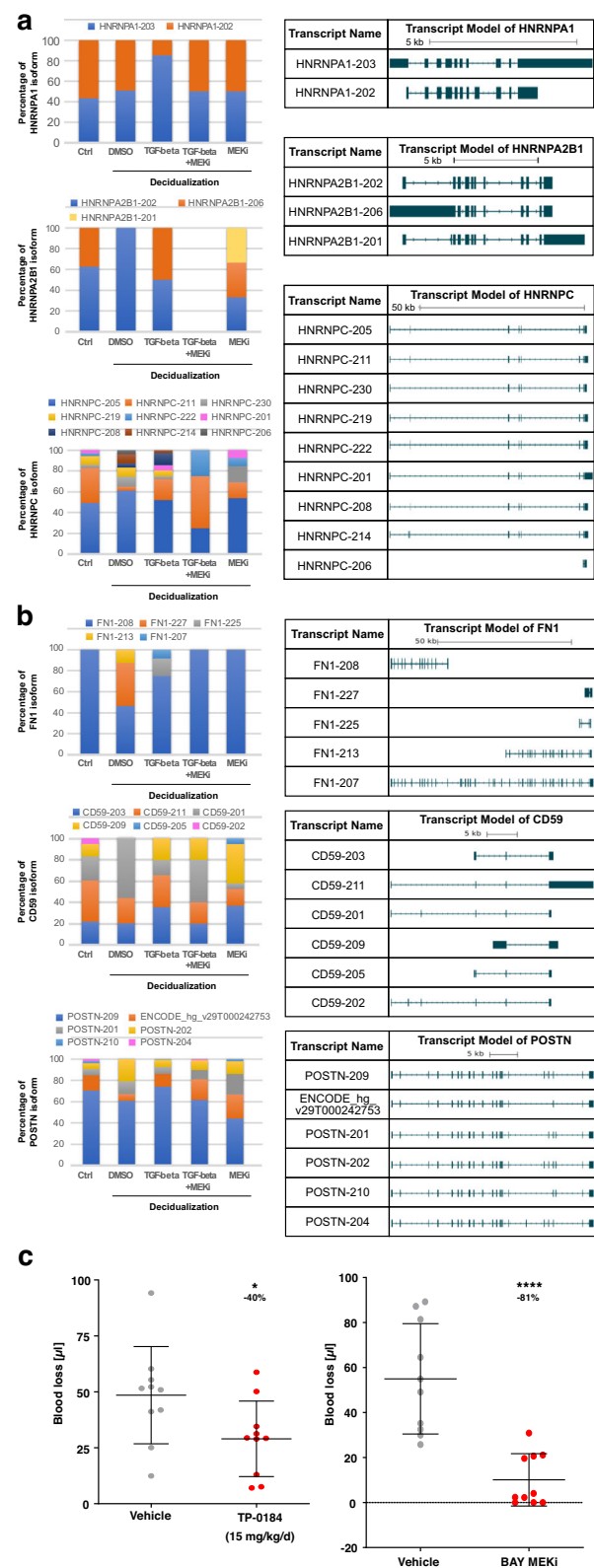

**Fig. 5 | The effect of TGF-β on endometrial homoeostasis in vitro and in vivo.**
**a**, **b** Alternative transcript usage induced by TGF-β treatment in decidualized THESC cell line. **a** Members of the heterogeneous nuclear ribonucleoprotein (hnRNP) family: *HNRNPA1*, *HNRNPA2B1*, and *HNRNPC*. **b** ECM-related genes (*FN1*, *POSTN*) and the immune-related gene *CD59*. Sample size per group is 3. **c** Effect of TP-0184 (ACVR inhibitor; left panel) and BAY MEKi (MEK inhibitor; right panel) on menstrual-like bleeding in a murine model. Two independent in vivo experiments were conducted to investigate the effects of TP-0184 (an ACVR inhibitor) and BAY-533 (a MEK inhibitor); the sample size for each group is 10. Both treatments showed a significant reduction in total uterine blood loss. Blood loss was quantified via alkaline elution of tampons and corrected for background levels. Data represents the mean with standard deviation from ten experiments per treatment group. Statistical significance was assessed using Student's *t*-test (*$p < 0.05$; ****$p < 0.0001$).

Supporting this, our multi-omics analysis confirmed upregulation of key ECM components, including *COL1A1, COL3A1* and *VCAN*, consistent with previous studies demonstrating excessive collagen synthesis in UFs[25,94,97–100]. Collagen fibrils for example, were found shorter and more disordered in UFs, in addition to the altered ratio of collagen type I/III[101]. Moreover, sc-RNAseq revealed elevated receptor-ligand interactions in collagen, laminin, and fibronectin-1 signalling in UF endometrium and myometrium, suggesting that ECM remodelling extends beyond fibroid itself to the surrounding uterine tissues[31,102]. These findings reinforce the hypothesis that targeting ECM-related pathways may offer therapeutic potential in treating UF and its associated symptoms[94,95,103–107].

Our study also highlights RNA processing and alternative splicing as critical contributors to endometrial dysfunction in UF patients. Alternative splicing plays a crucial role in protein diversity and has been linked to various diseases, including cancer[108–110]. Our multi-omic analysis identified latent factors that correlates with HMB, hormone treatment, and fibroid presence with certain driver mutations, emphasizing the broad impact of UF on endometrial physiology. We found that RNA metabolic processes and splicing-related genes were noticeably dysregulated, implicating that aberrant transcript usage may contribute to UF-associated HMB.

Further DTU analysis revealed alternative splicing in genes involved in blood vessel morphogenesis (*TGFBR2*, *ENG*, and *NRP1*), prostaglandin synthesis (*TBXA2R*, *PTGES*), and hormone signalling (*PGR*, *FGF7* and *FGFR2*). DTU in splicing-related genes (*HNRNPR*, *HNRNPL*) further underscores the potential disruption of splicing regulation in UF-associated endometrial pathology.

Notably, the TGF-β type II receptor emerged as a key regulator, with an altered balance between its two isoforms, TβR-II and TβRII-B, which binds TGF-β I/III or TGF-β II, respectively[72,73]. Our findings suggest a shift toward the dominant expression of TβRII-B in UF endometrium, potentially influencing TGF-β ligand specificity and downstream signalling effects. These findings suggest that alternative splicing in UF endometrium may alter TGF-β signalling dynamics, further compromising endometrial tissue homoeostasis, ECM remodelling and fibrotic processes.

TGF-β signalling[111,112] is a known regulator of alternative splicing, acting through pathways such as SMAD and PI3K/Akt/SRPK1[113–118] to influence exon inclusion and exclusion. Our sc-RNAseq analysis revealed TGF-β signalling is strikingly upregulated in UF endometrium, with elevated expression of TGF-β receptors. Given the well-established elevation of TGF-β levels in UF tissues, our findings suggest that fibroids may serve as a source of TGF-β ligands, which in turn influence alternative splicing and transcript expression profile in endometrium.

To validate the role of alternative splicing in endometrial physiology, we examined transcript isoform changes in vitro using TGF-β treated THESC cells during decidualization. Long-read sequencing analysis identified DTU in genes regulating RNA splicing including hnRNP family, *RBM4* and *RBM39*, ECM organisation like *FN1*, *POSTN*, and immune response like *CD59*[119–124]. We showed a shift in isoform ratios for *HNRNP* genes, *FN1*, and *POSTN*, suggesting that TGF-β signalling may affect ECM

insertion-deletion and frameshift variants. Given that ECM dysregulation is a hallmark of UF, the identification of *COL4A6* variants further underscores the functional impacts on ECM remodelling. In addition to hormone regulation, key mechanisms that contribute to ECM remodelling in UFs include Rho and ERK/p38 related mechanotransduction, nuclear location of YAP/TAZ, and growth factors such as TGF-β, EGF, and IGF-1[94–96].

remodelling and UF progression via directly influencing alternative splicing factors and subsequent splicing events.

The MEK/ERK (MAPK) pathway plays a critical role in uterine fibroid pathophysiology, particularly in mediating fibroid cell proliferation, extracellular matrix (ECM) deposition, and angiogenesis, all of which contribute to heavy menstrual bleeding (HMB). Several studies have demonstrated that growth factors highly expressed in fibroids, such as TGF-β, IGF, and PDGF, activate the MEK/ERK pathway, driving fibrotic and angiogenic changes that disrupt endometrial homoeostasis[83,84,95,125]. Our findings that blocking the TGF-β or MEK signalling cascade in murine menstruation model reduced blood loss indicates TGF-β-driven changes, particularly those affecting the ECM and vasculature, contribute to HMB through an ERK-dependent pathway.

Our findings have noticeable implications for potential UF treatment strategies. Given that ECM stiffness has been linked to alternative splicing through activation of Ser/Arg-rich spliceosome proteins[126], targeting ECM remodelling and TGF-β-mediated splicing regulation may provide potential therapeutic avenues. Current antifibrotic approaches, such as collagenase treatment or inhibition of fibrotic gene expression, have been shown to reduce ECM density and fibroid cell proliferation[94]. Moreover, compounds such as epigallocatechin gallate (EGCG) from green tea, have been shown to reduce fibroid volume and improve HMB, potentially through targeting on fibrotic signalling pathways including TGF-β, β-catenin, JNK and AKT pathways which are involved in fibrotic progression[127]. Further studies should further explore the therapeutic potential of splicing modulators and antifibrotic agents in mitigating UF progression and associated symptoms.

## Data availability

All raw and processed sequencing data associated with Figs. 1–5 and all Supplementary Figs. in this study are available in the NCBI's Gene Expression Omnibus: bulk RNA Sequencing data (GSE199849) and single-cell RNA sequencing data (GSE220650) of patient samples applied to this study; Illumina short-read and ONT long-read RNA sequencing of in vitro THESC decidualization (GSE261366). The mass spectrometry proteomics data have been deposited to the ProteomeXchange Consortium via the PRIDE[128] partner repository with the dataset identifier PXD051220. The source data for the graphs in Figs. 1–5 in the main manuscript can be found in the Supplementary Data 6.

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

## Acknowledgements

This work was supported through the Bayer - Oxford Alliance in Women's Healthcare, which receives funding through the NIHR Biomedical Research Centre, the Endometriosis CaRe Centre Oxford, Oxford University Medical Sciences Division and Bayer Healthcare. Further research support was obtained from Innovate UK (UO, MP, APC), the National Institute for Health Research Oxford Biomedical Research Centre (UO), Cancer Research UK (CRUK, UO), the Bone Cancer Research Trust (APC and UO), the Leducq Epigenetics of Atherosclerosis Network (LEAN) programme grant from the Leducq Foundation (UO), the Chan Zuckerberg Initiative (APC) and the Myeloma Single Cell Consortium (UO). APC is a recipient of an MRC Career Development Fellowship (MR/V010182/1). Work in the BMK laboratory was supported by the Wellcome Trust (097812/Z/11/Z) and the Engineering and Physical Science Research Council (EP/N034295/1).

## Author contributions

U.O., CY.W., M.P. and A.P.C. designed and supervised the study; C.Y.W., M.P. and U.O. wrote the first manuscript draft. CY.W., A.P.C. and U.O. revised the draft versions of the manuscript. K.Z., C.M.B., J.M. (Oxford), K.G., S.M., M.M. supervised and performed sample collection and clinical annotation, with important help from C.M.B., T.M.Z. and A.L.H., C.Y.W., M.P., D.O.B., J.M. (Oxford), N.M., V.G., B.M., S.B., R.F. performed experiments. C.Y.W., D.O.B., A.P.C., J.M. (Bayer) performed data analysis, with significant contributions from A.N., M.O., B.K., and A.L.H. C.M.B., K.Z., A.L.H., S.M., J.M. (Bayer), N.S. and T.M.Z. contributed critical data interpretation. All authors have read and provided input to the manuscript.

## Competing interests

FS, MO, NS, JM, and TMZ are employees and shareholders of Bayer Pharmaceuticals. MP, APC and UO are co-founders of Caeruleus Genomics plc. The study was jointly supported by Oxford and Bayer Healthcare; conceptualisation, research, data analysis and presentation were conducted in an unbiased manner and not influenced by the funding bodies.

## Additional information

[1]Botnar Research Centre, NIHR BRC, University of Oxford, Oxford, UK. [2]Target Discovery Institute, Centre for Medicines Discovery, Nuffield Department of Medicine, University of Oxford, Oxford, UK. [3]Nuffield Department of Women's & Reproductive Health, University of Oxford, Oxford, UK. [4]Department of Oncology, University of Oxford, Oxford, UK. [5]Research and Early Development, Bayer AG, Berlin, Germany. [6]These authors contributed equally: Chen-Yi Wang, Martin Philpott. ✉e-mail: Udo.oppermann@ndorms.ox.ac.uk; Adam.cribbs@ndorms.ox.ac.uk

