## [Transparent Peer Review file · Communications Medicine]

A systems-based approach to uterine fibroids identifies differential splicing associated with abnormal uterine bleeding

Corresponding Author: Professor Adam Cribbs

Version 0:

Reviewer comments:

Reviewer #1

(Remarks to the Author)

This extensive study addresses a significant concern associated with uterine fibroids, utilizing a range of advanced techniques, which is highly commendable. However, the study primarily reiterates findings already reported, such as the involvement of extracellular matrix components, growth factors like TGF- β , and angiogenesis, thus lacking in novelty. Additionally, the direct mechanisms linking these factors to heavy menstrual bleeding (HMB) remain unclear or not clearly interpreted. Additionally, the mouse model used in the study is not out of concern. Despite these points, the in vivo data show promise and have the potential to be translated into human studies.

The introduction is precisely described.

The discussion section needs to be improved.

Fibroid Sample Inclusion: What part of the fibroid was included in the study? Was it from the center, or did it include the pseudocapsule (outer part of the fibroid)?

Results Section Clarifications:

Genomic Analysis Section: In the section titled "Genomic analysis suggests multiple lesions and novel candidates and pathways in UF Pathology," could you please elaborate on the novel candidates and pathways discussed?

IL-11 Levels: The data indicate that IL-11 levels were upregulated in patients with HMB. Did you study IL-11 in the mouse model of HMB?

Supplementary Figure 9: In the sentence "(Supplementary Fig. 9), aligning with our previous analyses," could you add a reference or comment on this?

Extracellular Matrix Components: The sentence "Additionally, among the features positively associated with fibroid, a predominant number were components of the ECM, including structural proteoglycans like versican (VCAN) and collagen proteins (Fig. 3e and Supplementary Fig. 10)" is consistent with a recent study (PMID: 37231028). Could you comment on this?

Pathway Analysis: The pathway analysis revealed that features in factor 2 and factor 7 were predominantly involved in RNA processing and metabolic processes, including mRNA splicing and RNA 3'-end processing (Fig. 4d, middle and bottom panels, 4f-g; Supplementary Fig. 16-17). These data also indicate a positive relation of factors like YAP1 with fibroid occurrence. The role of YAP in fibroid growth has been reported earlier (PMID: 34323413; PMID: 37231028). Could you comment on this?

Steroid Hormones: Since it is well established that steroid hormones impact fibroid growth, the involvement of steroid hormones and their receptors are either lacking or not clearly interpreted in the study. Please comment on this.

MEK Inhibitor Clarification: No definitive conclusions regarding the involvement of MEK with HMB were addressed before the section "A MAPK/ERK kinase (MAP2K, MEK) inhibitor (MEKi)61 was applied to probe the effect of TGF- β by blocking the downstream signaling cascade." Could you explain the rationale behind using MEKi?

Figure 7 Title:

The title "Transcript usage dynamics during TGF- β induced in vitro decidualization, as revealed by long read sequencing" does not represent the data presented in the figure. Please clarify how this relates to "Illustrates the impact of MEK inhibitor (MEKi, left panel) or ACVR inhibitor (TP-0184, right panel) treatments on menstrual-like bleeding in a murine model, showing a significant reduction in total uterine blood loss."

Figure Adjustments:

Please adjust the texts in Fig. 7d. The left panel should indicate TP-0184 (ACVR inhibitor) and the right panel should indicate MEKi (MEK inhibitor).

Reviewer #2

(Remarks to the Author)

The study of Wang and colleagues is a very interesting investigation on uterine fibroids. The authors analysed myometrial, fibroids and endometrium tissue from 91 patients performing genetic, transcriptional and proteomic approaches. Overall the study furnishes novel information regarding uterine fibroids biology. The second part of the study using in vitro and in vivo study is less straightforward, coherent and not well integrated and connected to the study. The information regarding age and other pathological conditions of the patients are missing.

Reviewer #3

(Remarks to the Author)

The manuscript by Wang and coworkers investigates in a systematic approach the changes in the myometrium of patients with uterine fibroids. This is a data intense manuscript that looks at a large number of patients and investigates mutations in key leiomyoma, changes in the transcriptome and proteome in a multiomic approach. The strength of this manuscript is in the large amount of data presented that will serve for investigators to mine. Aside from the large amount of data there are several weaknesses

This manuscript needs to be rewritten to limit the reliance on supplemental data in the text and focus on the main figures. The supplemental figures are that supplemental. If they are key to the findings they should be included in the manuscript. Also more explanation is needed. For example it lists 10 factors from their multinomic analysis but does not define what they mean by these 10 factors.

The weakness is in the integration of the endometrial data. While this manuscript looks at differences between UF patients and controls and compares changes in the myometrium and endometrium there is no clear synthesis of interactions. It is unclear why this manuscript investigated the endometrium and made no serious effort to integrate this data with the UF data. The data on the endometrium should be removed. And published as a separate study.

One finding that is interesting in this manuscript is the role of TGF beta signaling in splicing. However even though this manuscript focus is on the myometrium, it validates the role of TGF β in endometrial stroma cells. This should be one in myometrial cell lines.

Again there is a lot of data in this manuscript but no clear focus on mechanism or integration.

Version 1:

Reviewer comments:

Reviewer #1

(Remarks to the Author)

The authors made substantial changes and significantly improved the manuscript.

Reviewer #3

(Remarks to the Author)

The authors have sufficiently addressed the issues of the initial review as best they can.

Reviewer #1 (Remarks to the Author):

R1.1: This extensive study addresses a significant concern associated with uterine fibroids, utilizing a range of advanced techniques, which is highly commendable. However, the study primarily reiterates findings already reported, such as the involvement of extracellular matrix components, growth factors like TGF- β , and angiogenesis, thus lacking in novelty. Additionally, the direct mechanisms linking these factors to heavy menstrual bleeding (HMB) remain unclear or not clearly interpreted. Additionally, the mouse model used in the study is not out of concern. Despite these points, the in vivo data show promise and have the potential to be translated into human studies.

Response R1.1: We now emphasise the novelty of our findings by demonstrating how alternative splicing events, TGF- β signalling, and extracellular matrix remodelling contribute to uterine fibroid (UF)-associated endometrial dysfunction, particularly in the context of heavy menstrual bleeding (HMB). Our study offers mechanistic insights into how these molecular alterations disrupt endometrial homeostasis, establishing a direct link to HMB. Furthermore, we have clarified the rationale for our mouse model, detailing its relevance to human pathophysiology. Throughout the manuscript, we have implemented extensive revisions to enhance the clarity and accessibility of our results to the reader.

R1.2: The introduction is precisely described.

Response R1.2: Many thanks for this comment.

R1.3: The discussion section needs to be improved.

Response R1.3 (section DISCUSSION): We have substantially modified and improved the discussion.

R1.4: Fibroid Sample Inclusion: What part of the fibroid was included in the study? Was it from the center, or did it include the pseudocapsule (outer part of the fibroid)?

Response R1.4 (Line 509-511): The majority of the fibroid samples analysed were from the central region. However, pseudocapsule tissue was available in a limited number of patients, and where present, it was included in the study. We acknowledge the potential biological significance of the pseudocapsule and have noted this as a consideration for future investigations.

R1.5: Genomic Analysis Section: In the section titled “Genomic analysis suggests multiple lesions and novel candidates and pathways in UF Pathology,” could you please elaborate on the novel candidates and pathways discussed?

Response R1.5: This section has been removed from the revised manuscript to improve focus and clarity. However, key findings related to novel candidates and pathways have been integrated into the relevant results sections to ensure a more precise and contextual presentation of the data.

R1.6: IL-11 Levels: The data indicate that IL-11 levels were upregulated in patients with HMB. Did you study IL-11 in the mouse model of HMB?

Response R1.6 (Line 131-133): We acknowledge the relevance of IL-11 in HMB and its potential as a therapeutic target. However, IL-11 was not specifically studied in our mouse model. Notably, IL-11 has been evaluated in a phase II clinical trial, where recombinant IL-11

(rhIL-11, Neumega®) was administered subcutaneously for up to 7 days during the menstrual cycle over six months (doi: 10.1160/TH11-04-0274). The final report (2018) demonstrated a >50% reduction in menstrual blood loss, as assessed by the Pictorial Blood Assessment Chart (PBAC) (ClinicalTrials.gov ID: NCT00524342). These findings support the therapeutic potential of IL-11 in HMB, warranting further investigation into its mechanistic role in UF-associated HMB.

R1.7: Supplementary Figure 9: In the sentence “(Supplementary Fig. 9), aligning with our previous analyses,” could you add a reference or comment on this?

Response R1.7 (Line 161-164): The original Supplementary Figure 9 has been removed from the revised manuscript to improve clarity and streamline the presentation of results. Consequently, the referenced statement has been adjusted accordingly.

R1.8: Extracellular Matrix Components: The sentence “Additionally, among the features positively associated with fibroid, a predominant number were components of the ECM, including structural proteoglycans like versican (VCAN) and collagen proteins (Fig. 3e and Supplementary Fig. 10)” is consistent with a recent study (PMID: 37231028). Could you comment on this?

Response R1.8 (Line 324-332, ref 73): We have addressed this in the Discussion section (Reference 73), highlighting how our findings align with and expand upon the recent study (PMID: 37231028). Specifically, we discuss the role of extracellular matrix components such as collagen proteins in fibroid pathophysiology and how our data contribute to a broader understanding of ECM remodelling in uterine fibroids.

R1.9: Pathway Analysis: The pathway analysis revealed that features in factor 2 and factor 7 were predominantly involved in RNA processing and metabolic processes, including mRNA splicing and RNA 3'-end processing (Fig. 4d, middle and bottom panels, 4f-g; Supplementary Fig. 16-17). These data also indicate a positive relation of factors like YAP1 with fibroid occurrence. The role of YAP in fibroid growth has been reported earlier (PMID: 34323413; PMID: 37231028). Could you comment on this?

Response R1.9 (Line 321-323 and Line 377-382; ref 73 and 106): We have addressed this in the Discussion section (References 73, 106), contextualising our findings within existing literature (PMID: 34323413; PMID: 37231028). Specifically, we discuss how our pathway analysis highlights RNA processing pathways, including mRNA splicing and RNA 3'-end processing, as key features associated with fibroid pathology. Additionally, we comment on the key mechanisms including on ECM remodelling in fibroid growth, aligning with previous studies while providing new insights into its potential mechanistic contributions.

R1.10: Steroid Hormones: Since it is well established that steroid hormones impact fibroid growth, the involvement of steroid hormones and their receptors are either lacking or not clearly interpreted in the study. Please comment on this.

Response R1.10 (Line 302-306): We have addressed the role of steroid hormones and their receptors in the Discussion section, specifically in the first paragraph. In this revision, we comment on how steroid hormone-related treatments influence heavy menstrual bleeding (HMB) and fibroid progression, incorporating relevant literature to contextualise our findings within the broader understanding of fibroid pathophysiology.

R1.11: MEK Inhibitor Clarification: No definitive conclusions regarding the involvement of MEK with HMB were addressed before the section “A MAPK/ERK kinase (MAP2K, MEK) inhibitor (MEKi)61 was applied to probe the effect of TGF- β by blocking the downstream signaling cascade.” Could you explain the rationale behind using MEKi?

Response R1.11 (Line 251-258): We acknowledge the need for further clarification on the rationale for using a MEK inhibitor (MEKi) in our study. The MEK/ERK (MAPK) pathway plays a critical role in uterine fibroid pathophysiology, particularly in mediating fibroid cell proliferation, extracellular matrix (ECM) deposition, and angiogenesis—all of which contribute to heavy menstrual bleeding (HMB). Several studies have demonstrated that growth factors highly expressed in fibroids, such as TGF- β , IGF, and PDGF, activate the MEK/ERK pathway, driving fibrotic and angiogenic changes that disrupt endometrial homeostasis (Borahay *et al.*, 2015; Reschke *et al.*, 2022; Ciarmela *et al.*, 2011).

TGF- β , in particular, is a key driver of ECM remodelling in fibroids, and its fibrotic effects are mediated through both Smad-dependent and non-canonical MEK/ERK signaling pathways (Salama *et al.*, 2012; Uimari *et al.*, 2022). Prior work has shown that blocking MEK/ERK can attenuate fibroid cell proliferation and ECM production, suggesting that ERK activation is required for certain TGF- β -mediated effects in fibroid pathology (Ciarmela *et al.*, 2011; Reschke *et al.*, 2022). Given that aberrant ECM accumulation and dysregulated angiogenesis are key contributors to fibroid-associated HMB, we applied a MEK inhibitor to determine whether TGF- β -mediated signalling relevant to these processes was dependent on MEK/ERK activation.

Our approach is supported by previous studies demonstrating that inhibiting MEK/ERK in fibroid cells reduces collagen deposition, angiogenic factor secretion, and proliferation (Borahay *et al.*, 2015; Reschke *et al.*, 2022). The use of MEKi in this study provides mechanistic insight into whether TGF- β -driven changes in the endometrium homeostasis, particularly those affecting the ECM and vasculature, contribute to HMB through an ERK-dependent pathway.

References:

- Borahay MA *et al.* (2015). *Mol Med*, 21(1):242–256. PMID: 25879625.
- Ciarmela P *et al.* (2011). *Hum Reprod Update*, 17(6):772–790. PMID: 21788281.
- Reschke L *et al.* (2022). *F&S Science*, 3(4):383–391. PMID: 35598777.
- Salama SA *et al.* (2012). *Fertil Steril*, 98(1):178–184. PMID: 22579131.
- Uimari *et al.* (2022). *Front Reprod Health*, 4:839243. PMID: 36303616.

R1.12: Figure 7 Title: The title “Transcript usage dynamics during TGF- β induced in vitro decidualization, as revealed by long read sequencing” does not represent the data presented in the figure. Please clarify how this relates to “Illustrates the impact of MEK inhibitor (MEKi, left panel) or ACVR inhibitor (TP-0184, right panel) treatments on menstrual-like bleeding in a murine model, showing a significant reduction in total uterine blood loss.”

Response R1.12 (Figure 5): We appreciate the reviewer’s feedback and have revised the title of Figure 5 to better align with the data presented. The updated title now explicitly reflects the effect of TGF- β on endometrial homeostasis *in vitro* and *in vivo*. The impact of MEK inhibitor (MEKi) and ACVR inhibitor (TP-0184) treatments on menstrual-like bleeding in a murine model demonstrate the potential impact of TGF- β , ensuring consistency with the figure content. Additionally, we have clarified the link between TGF- β signalling, transcript usage, and *in vivo* functional validation, reinforcing the rationale for targeting MEK and ACVR pathways in UF-associated HMB.

R1.13: Please adjust the texts in Fig. 7d. The left panel should indicate TP-0184 (ACVR inhibitor) and the right panel should indicate MEKi (MEK inhibitor).

Response R1.13 (Figure 5): Thanks for pointing this out. We have modified our figure accordingly and this is now labelled as Fig. 5.

Reviewer #2 (Remarks to the Author):

R2.1: The study of Wang and colleagues is a very interesting investigation on uterine fibroids. The authors analysed myometrial, fibroids and endometrium tissue from 91 patients performing genetic, transcriptional and proteomic approaches. Overall the study furnishes novel information regarding uterine fibroids biology. The second part of the study using *in vitro* and *in vivo* study is less straightforward, coherent and not well integrated and connected to the study.

Response R2.1: We have made extensive modifications to our manuscript by focusing on the impact of uterine fibroids (UFs) on UF-associated endometrial dysfunction. This integration is essential to understanding how fibroids contribute to heavy menstrual bleeding (HMB) and other pathological changes in the endometrium. The second part of the *in vitro* and *in vivo* study highlights TGF- β signalling on endometrium homeostasis.

R2.2: The information regarding age and other pathological conditions of the patients are missing.

Response R2.2 (Supplementary Table 1): We have included the information on age and other pathological conditions of the patients in Supplementary Table 1. This additional data provides relevant clinical context and enhances the transparency of our cohort characteristics.

Reviewer #3 (Remarks to the Author):

R3.1: The manuscript by Wang and coworkers investigates in a systematic approach the changes in the myometrium of patients with uterine fibroids. This is a data intense manuscript that looks at a large number of patients and investigates mutations in key leiomyoma, changes in the transcriptome and proteome in a multi-omic approach. The strength of this manuscript is in the large amount of data presented that will serve for investigators to mine. Aside from the large amount of data there are several weaknesses

Response R3.1: We appreciate the reviewer's acknowledgment of the dataset's value. In the revised manuscript, we have significantly strengthened the mechanistic focus by improving the integration of multi-omic data, ensuring clearer links between genomic alterations, transcriptomic changes, and proteomic signatures in fibroids and associated endometrial dysfunction. We have also refined the presentation of key findings to enhance clarity and provide a more structured synthesis of UF pathophysiology. These revisions aim to improve the interpretability and impact of the data for future investigations.

R3.2: This manuscript needs to be rewritten to limit the reliance on supplemental data in the text and focus on the main figures. The supplemental figures are that supplemental. If they are key to the findings they should be included in the manuscript.

Response R3.2: We have revised the manuscript to reduce reliance on supplemental data in the main text, ensuring that key findings are presented within the primary figures. Most supplementary figures have been removed, retaining only those that provide essential supportive information. This restructuring enhances the clarity and accessibility of our findings within the main manuscript.

R3.3: Also more explanation is needed. For example it lists 10 factors from their multi-omic analysis but does not define what they mean by these 10 factors.

Response R3.3 (Line 136-142, 654-657,661-667): We have expanded the explanation of the latent factors identified in our multi-omic analysis, providing a clear definition of their significance and biological relevance. Additionally, we have examined their correlation with known clinical, biological and technical variables to contextualise their potential impact. These clarifications have been incorporated into the Results and Methods section to improve interpretability.

R3.4: The weakness is in the integration of the endometrial data. While this manuscript looks at differences between UF patients and controls and compares changes in the myometrium and endometrium there is no clear synthesis of interactions.

Response R3.4 (Line 136-149, 155-159, 187-210): We have significantly improved the integration of endometrial data by explicitly linking UF-associated genetic and transcriptomic alterations to downstream effects on endometrial function. In the revised manuscript, we now provide a more cohesive synthesis of interactions between fibroid pathology and endometrial dysfunction, particularly highlighting how TGF- β signalling, ECM remodelling, and alternative splicing contribute to heavy menstrual bleeding (HMB). Additionally, we have strengthened the comparative analyses, illustrating how fibroid-driven molecular changes impact endometrial homeostasis. These refinements provide a clearer mechanistic framework for understanding UF-associated endometrial dysfunction.

R3.5: It is unclear why this manuscript investigated the endometrium and made no serious effort to integrate this data with the UF data. The data on the endometrium should be removed. And published as a separate study.

Response R3.5 (Line 136-149, 155-159, 187-210, 223-240): We have clarified the rationale for including endometrial data by focusing on the impact of uterine fibroids (UFs) on UF-associated endometrial dysfunction. This integration is essential to understanding how fibroids contribute to heavy menstrual bleeding (HMB) and other pathological changes in the endometrium. The revised manuscript strengthens this connection by explicitly linking UF-driven molecular changes to endometrial dysfunction.

R3.6: One finding that is interesting in this manuscript is the role of TGF beta signaling in splicing. However even though this manuscript focus is on the myometrium, it validates the role of TGFb in endometrial stroma cells. This should be one in myometrial cell lines.

Response R3.6: We have focused on the impact of uterine fibroids (UFs) on UF-associated endometrial dysfunction and have identified that UFs influence endometrial function through mRNA splicing and alternative transcript usage, potentially mediated by TGF- β signalling and ECM stiffness. To examine transcript usage changes in the endometrium due to TGF- β signalling, we used THESC (human endometrial stromal cells) for validation. While our primary focus remains on the myometrium, these findings highlight a potential fibroid-endometrium interaction that warrants further investigation in myometrial cell lines in future studies.

R3.7: Again there is a lot of data in this manuscript but no clear focus on mechanism or integration.

Response R3.7: We have substantially revised the manuscript to enhance clarity and focus. Specifically, we have refined the integration of multi-omic data to elucidate the mechanistic role of uterine fibroids (UFs) in endometrial dysfunction and heavy menstrual bleeding (HMB). The revised version provides a more structured narrative that explicitly links genomic alterations, transcriptomic changes, and proteomic findings to key pathological mechanisms, including TGF- β signalling, ECM dysregulation, and alternative splicing. This refinement strengthens the biological interpretation and ensures a clearer mechanistic insight into UF-associated endometrial dysfunction.